# GRASP-GS: GEOMETRIC REGISTRATION AND DUAL-STAGE SALIENCY PRUNING FOR EFFICIENT 3D GAUSSIAN SPLATTING

## ABSTRACT

3D Gaussian Splatting (3DGS) has emerged as a powerful paradigm for scene representation, enabling real-time rendering with high visual fidelity by modeling scenes as anisotropic 3D Gaussians. However, existing methods suffer from blurred reconstructions, redundant Gaussians, and high training costs due to sparse Structure-from-Motion (SfM) initialization and heuristic densification. In this paper, we propose GRASP-GS (**G**eometric **R**egistration and Du**A**l-Stage **S**aliency **P**runing for 3D **G**aussian **S**platting), a framework that integrates geometric prior-guided initialization with adaptive saliency pruning. Our method first enhances the initial point cloud by extracting and fusing dense multi-view features with the SfM points through multi-stage refinement. Then, a dual-stage pruning strategy sequentially applies KL-based Rendering Survival Pruning (KL-RSP) to reduce spatial redundancy and Opacity-based Density-Constrained Pruning (ODCP) to eliminate low-contribution Gaussians. Experiments demonstrate that GRASP-GS achieves compact and high-quality scene representations, enabling efficient real-time rendering with enhanced structural integrity and visual quality.

## 1 INTRODUCTION

Novel View Synthesis (NVS) aims to generate photorealistic images from arbitrary viewpoints using sparse input observations. It is a cornerstone of 3D reconstruction and rendering, with applications in virtual reality(Fei et al., 2024), autonomous driving(Zhou et al., 2024), and digital twins(Wang et al., 2024). Neural Radiance Fields (NeRF)(Mildenhall et al., 2021) achieve impressive visual quality by implicitly modeling scenes with MLPs, but their high computational cost and slow rendering limit real-time use.

3D Gaussian Splatting (3DGS)(Kerbl et al., 2023) offers a real-time alternative by representing scenes as collections of anisotropic 3D Gaussians with differentiable rasterization(Chen & Wang, 2024). However, 3DGS still faces challenges in complex or large-scale scenes due to sparse Structure-from-Motion (SfM) point clouds and heuristic densification, which often introduce redundant Gaussians and elevate training cost(Deng et al., 2025; Foroutan et al., 2024; Grubert et al., 2025). This highlights a trade-off between rendering quality, geometric accuracy, and efficiency, motivating methods that jointly improve initialization and density control.

Prior work addresses these aspects separately. For initialization, Depth-Reg(Chung et al., 2024), GP-GS(Guo et al., 2025), Point Cloud Densification(Chan et al., 2024), and GaussianPro(Cheng et al., 2024) leverage depth priors to increase geometric completeness, but suffer from depth noise and scale inconsistencies. For density control, AbsGS(Ye et al., 2024), EDC(Deng et al., 2025), Mini-Splatting(Fang & Wang, 2024b), Mini-Splatting2(Fang & Wang, 2024a), LightGaussian(Fan et al., 2023), and RadSplat(Niemeyer et al., 2024) employ gradient- or contribution-based pruning to reduce redundancy. These solutions often neglect the interplay between initialization and pruning, limiting overall effectiveness.

To address this gap, we propose GRASP-GS, a modular yet integrated framework for 3DGS that combines geometric prior-guided initialization with dual-stage saliency pruning (Figure 1). GRASP-GS can be seamlessly integrated into existing pipelines, enhancing initialization quality and reducing redundancy. Better initialization reduces the burden on density control, while adaptive pruning

Figure 1: **Overview of GRASP-GS.** Multi-view images are processed by MoGe to extract affine-invariant dense 3D points. These points are first filtered through geometric-chromatic consistency with sparse SfM points, then refined and consolidated to form a clean and structured dense prior. The fused points are used to initialize 3D Gaussians, which are optimized via iterative densification and dual-stage saliency pruning: KL-based Rendering Survival Pruning (KL-RSP) and Opacity-based Density-Constrained Pruning (ODCP).

maintains and refines the geometric structure from initialization rather than degrading it, resulting in compact yet faithful scene representations.

The main contributions are as follows:

- **Geometric Prior-Guided Initialization**: Affine-invariant monocular priors with multi-stage refinement improve completeness and accuracy of initial 3DGS geometry.
- **Dual-Stage Saliency Pruning**: KL-based Rendering Survival Pruning combined with opacity-based density filtering effectively removes redundancy while preserving structure.
- **Integrated yet Modular Framework**: GRASP-GS jointly enhances initialization and pruning, improving rendering quality and model compactness while remaining compatible with baseline 3DGS pipelines.

## 2 RELATED WORKS

3D Gaussian Splatting (3DGS) (Kerbl et al., 2023) represents scenes explicitly using anisotropic Gaussian primitives with differentiable rasterization, enabling real-time rendering with high visual fidelity. This overcomes the computational limitations of Neural Radiance Fields (NeRF) (Mildenhall et al., 2021), which achieve high-quality novel view synthesis but suffer from slow training and limited real-time capability. However, 3DGS relies on sparse SfM (Schonberger & Frahm, 2016) point clouds and heuristic densification strategies, which can introduce redundant Gaussians (Chen & Wang, 2024; Deng et al., 2025; Foroutan et al., 2024; Grubert et al., 2025).

To improve initialization from sparse SfM,several methods leverage priors: Depth-Reg (Chung et al., 2024) and GP-GS (Guo et al., 2025) use monocular depth and probabilistic consistency, while Point Cloud Densification (Chan et al., 2024) and GaussianPro (Cheng et al., 2024) generate dense seeds from depth estimates (Birkl et al., 2023; Yang et al., 2024). They still suffer from noise and scale inconsistencies. Recent affine-invariant geometry priors such as MoGe (Wang et al., 2025c) improve robustness but require careful integration with 3DGS pipelines.

Existing density control focuses on gradient-based strategies and contribution-aware pruning. AbsGS (Ye et al., 2024) and EDC (Deng et al., 2025) use gradient-aware splitting and axial pruning. Mini-Splatting (Fang & Wang, 2024b) and Mini-Splatting2 (Fang & Wang, 2024a) accelerate Gaussian simplification and densification. LightGaussian (Fan et al., 2023) and RadSplat (Niemeyer et al., 2024) prune low-contribution Gaussians based on rendering importance.Yang et al. (2025) localize error-hotspots to jointly calibrate geometry and insert points. While effective individually, these methods treat initialization and pruning separately, missing the synergy between high-quality initialization and intelligent density control.

## 3 METHODOLOGY

Our framework, **GRASP-GS**, improves 3D Gaussian Splatting by explicitly boosting the quality and compactness of the initial point cloud. As shown in Figure 1, it comprises two complementary modules. First, the Geometric Prior-Guided Initialization filters dense priors via geometric and chromatic consistency, then refines them into structured, view-consistent Gaussians. Second, the Dual-Stage Saliency Pruning cuts redundancy through spatial-similarity and opacity-aware constraints. Their synergy forms a feedback loop: the initialization yields a high-quality geometric base that relieves later optimization, while KL-RSP preserves dominant Gaussians within clusters and ODCP prunes low-contribution ones, so the initial quality is retained rather than eroded. The approach is general and readily plugged into other 3D-GS variants.

### 3.1 PRELIMINARY

**3D Gaussian Splatting.** 3D Gaussian Splatting (3DGS) (Kerbl et al., 2023) is a point-based scene representation that models geometry and appearance using a set of anisotropic 3D Gaussian primitives. Each Gaussian $G_i$ is parameterized by a 3D center $\boldsymbol{\mu}_i$, a rotation matrix $R_i$, a scaling matrix $S_i$, an opacity $\alpha_i$, and spherical harmonic (SH) coefficients for view-dependent color emission. Its spatial extent and orientation are encoded by the covariance matrix $\boldsymbol{\Sigma}_i = R_i S_i S_i^\top R_i^\top$, enabling compact modeling of complex geometry. The 3DGS pipeline typically begins with a sparse point cloud and camera poses obtained from Structure-from-Motion (SfM) systems (e.g., COLMAP). These inputs are used to initialize the Gaussians, which are then optimized through gradient descent to jointly refine geometric and appearance parameters. The optimization process is accompanied by structural operations such as Gaussian cloning, pruning, and transformation, aiming to improve rendering quality and computational efficiency. During rendering, each 3D Gaussian is projected onto the image plane as a 2D elliptical footprint and contributes to pixel colors through alpha compositing. The contribution of each Gaussian $G_i$ to a pixel is governed by its screen-space blending weight $w_i$, which depends on its projected opacity and relative depth. These weights, accumulated across camera rays, reflect the visibility and perceptual importance of each Gaussian.

**MoGe.** MoGe (Wang et al., 2025c) is a state-of-the-art monocular geometry estimation framework that directly predicts dense 3D point maps $P$ from single RGB images. Unlike conventional depth regression methods, MoGe estimates affine-invariant 3D points that satisfy the equivariance property: for any unknown global scale $s$ and translation $t$, the predicted geometry satisfies $P \approx sP + t$. This formulation removes focal length ambiguities and alleviates the need for metric supervision or precise intrinsics, making MoGe particularly robust in open-world scenes. To improve geometric fidelity, MoGe adopts a multilevel supervision strategy. At the global level, a rotation-optimized epipolar alignment (ROE) module leverages multiview camera poses to resolve scale and shift inconsistencies. Locally, a multiscale spherical surface fitting loss helps capture fine geometric structure, while auxiliary tasks such as normal estimation and semantic-aware masking encourage local planarity and suppress noise, especially in occluded or texture-less regions. MoGe serves as a dense, affine-consistent geometry prior. However, its raw outputs can be noisy and inconsistent across views, necessitating downstream registration and filtering for reliable 3D Gaussian initialization.

### 3.2 GEOMETRIC PRIOR-GUIDED INITIALIZATION

Our initialization framework consists of two stages designed to obtain a reliable geometric prior for 3D Gaussian Splatting. (1) **Geometric–Chromatic Filtering**: We select consistent viewpoints by jointly evaluating geometric alignment between dense monocular priors and sparse SfM reconstructions, as well as chromatic similarity across overlapping views. (2) **Refinement and Consolidation**: The filtered priors are further refined through outlier rejection and denoising, then aggregated into a unified point cloud representation. This two-stage process produces a robust and well-aligned prior that provides a stable starting point for 3D Gaussian initialization.

**Geometric-Chromatic Filtering.** To construct dense geometric priors, we first apply Farthest Point Sampling (FPS) (Moenning & Dodgson, 2003) on COLMAP-estimated camera poses to select 24 spatially diverse views. MoGe is then used to generate dense 3D point clouds from these selected images. For each candidate view, we establish correspondences by projecting MoGe points

into the image plane and checking their reprojection consistency with SfM keypoints; pairs with bidirectional reprojection errors below 2 pixels are considered valid matches. Using these correspondences, we perform a coarse-to-fine geometric registration that aligns dense priors with the global sparse point cloud (see Appendix A.1 for details). This alignment provides a reliable basis for computing two distance-based consistency measures:

- Geometric consistency $\mathcal{D}_{\text{geo}}$ measures the squared Euclidean distance between matched 3D points:

$$\mathcal{D}_{\text{geo}} = \frac{1}{M} \sum_{j=1}^{M} \|\mathbf{x}_j - \mathbf{y}_j\|^2 \tag{1}$$

- Chromatic consistency $\mathcal{D}_{\text{chro}}$ quantifies perceptual color differences in the CIE Lab color space:

$$\mathcal{D}_{\text{chro}} = \frac{1}{M} \sum_{j=1}^{M} \Delta E_{\text{CIE94}} \left( \mathbf{c}_j^{(1)}, \mathbf{c}_j^{(2)} \right) \tag{2}$$

The Lab color space is adopted due to its perceptual uniformity, where $\Delta E$ values correlate with human-visible differences. A chromatic threshold of 3.0 is used to represent a just-noticeable difference (Melgosa, 2016). A viewpoint is retained only if both $\mathcal{D}_{\text{geo}}$ and $\mathcal{D}_{\text{chro}}$ fall below their respective thresholds. To accommodate scene-level variation, the geometric threshold is adaptively updated using exponential moving averages. This filtering strategy ensures that only well-aligned and perceptually consistent priors contribute to the final reconstruction.

**Refinement and Consolidation.** To obtain a compact and reliable geometric prior for initializing GRASP-GS, the filtered multi-view point clouds are processed by a progressive refinement pipeline designed to suppress noise, reduce redundancy, and maintain structural fidelity. The pipeline consists of three stages.

First, to balance detail preservation and computational efficiency given the resolution-dependent density of MoGe outputs, we apply resolution-aware sampling. The number of retained points is proportional to the source image area:

$$N_{\text{sample}} = \max \left( \lfloor W \cdot H \cdot \rho \rfloor, N_{\text{min}}^{\text{pts}} \right), \tag{3}$$

where $W$ and $H$ denote the image width and height, $\rho$ is a density factor, and $N_{\text{min}}^{\text{pts}}$ ensures minimal coverage for low-resolution views. This prevents overfitting to high-resolution inputs while preserving geometric integrity.

Second, to remove residual noise—particularly in texture-less or occluded regions—we perform statistical denoising based on local point density. For each point $\mathbf{p}_i$, we compute the average distance to its $k$-nearest neighbors:

$$d_i = \frac{1}{k} \sum_{j \in \mathcal{N}_k(i)} \|\mathbf{p}_i - \mathbf{p}_j\|, \tag{4}$$

where $k$ is chosen to balance local sensitivity and robustness to sampling variations. We then discard points for which $d_i > \mu + \sigma$, with $\mu$ and $\sigma$ denoting the mean and standard deviation of all $d_i$—a moderate threshold that effectively removes isolated outliers while preserving fine geometric structures.

Finally, to enforce structural continuity and eliminate fragmented artifacts from view-dependent errors, we apply density-based clustering using DBSCAN (Ester et al., 1996), retaining only clusters with sufficient local support ($N_{\text{neighbors}} \geq N_{\text{min}}$), ensuring topological consistency across views.

After refinement, each point cloud is transformed using its estimated rigid alignment and fused into a global prior. The resulting representation is compact, denoised, and structurally consistent—providing a strong foundation for initializing 3D Gaussian optimization.

### 3.3 DUAL-STAGE SALIENCY PRUNING

To reduce redundancy while preserving both global structure and fine details, we propose a dual-stage saliency pruning framework. It comprises two complementary stages: (1) KL-based Rendering Survival Pruning (KL-RSP), and (2) Opacity-BasedDensity-Constrained Pruning (ODCP).

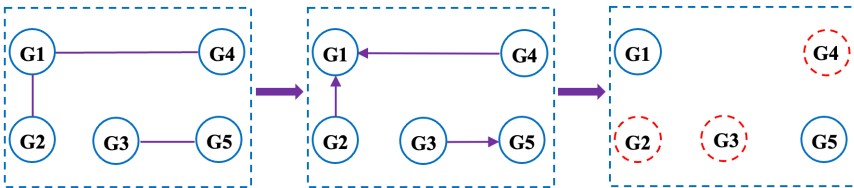

Figure 2: **KL-RSP via Union-Find Grouping.** Gaussians with pairwise KL divergence below the threshold $\tau_{\text{KL}}$ are clustered using a Union-Find structure. In each group, only the Gaussian with the highest rendering importance is preserved, while others highlighted with red circles are pruned to eliminate redundancy without degrading visual quality.

**KL-Based Rendering Survival Pruning (KL-RSP).** The core KL-RSP step measures redundancy between Gaussian splats using the Kullback–Leibler (KL) divergence, as formally defined in Appendix A.2. Each Gaussian $G_i$ is parameterized by mean $\mu_i$, covariance $\Sigma_i$, and appearance attributes $\{\mathbf{c}_i, \alpha_i\}$.

For efficiency, KL evaluation is restricted to a candidate subset of Gaussians obtained via gradient- and scale-based pre-filtering with voxel grouping (see Appendix A.2 for details).

A similarity graph is constructed where an edge $G_i \rightarrow G_j$ exists if $G_i$ is less important than $G_j$ within a KL-consistent pair. The importance is defined as

$$I(G_i) = \sum_{r=1}^{K} w_{ir}, \tag{5}$$

where $w_{ir}$ is the blending weight of $G_i$ along the $r$-th intersecting ray. Connected components are formed using Union-Find clustering (Galil & Italiano, 1991), and only the Gaussian with the highest importance in each component is retained. This KL-RSP step effectively removes redundant Gaussians while preserving perceptually dominant ones (Figure 2).

**Opacity-Based Density-Constrained Pruning (ODCP).** While KL-RSP effectively removes redundant Gaussians, it may leave behind isolated low-opacity splats that do not form KL-consistent clusters. To address this, we introduce opacity-based density-constrained pruning (ODCP). A Gaussian $G_i$ is pruned if its opacity $\alpha_i$ and local density $\rho_i$ satisfy:

$$\alpha_i < \tau_{\text{opacity}} \quad \text{and} \quad \rho_i > \tau_{\text{density}}, \tag{6}$$

where the local density is defined as

$$\rho_i = \frac{1}{\sqrt{\min_{j \neq i} \|\mu_i - \mu_j\|^2 + \epsilon}}. \tag{7}$$

The density threshold $\tau_{\text{density}}$ is dynamically set using quantile statistics, enabling adaptation to scene complexity. This complementary step suppresses visually negligible splats in dense regions while preserving detail in sparse areas.

Together, KL-RSP and ODCP form a dual-stage pruning strategy: At each pruning interval, KL-RSP is executed twice sequentially, followed by ODCP once, suppressing redundancy at both cluster and local-density levels and ensuring a compact yet faithful scene representation.

## 4 EXPERIMENTS

This section evaluates GRASP-GS through quantitative comparisons, qualitative analysis, and ablation studies.

### 4.1 EXPERIMENTAL SETUP

**Datasets.** Experiments are conducted on three benchmark datasets: **Mip-NeRF 360** (Barron et al., 2022) contains nine scenes (four indoor and five outdoor) with an average of 214 images per scene

| Metric \ Dataset / Method | Mip-NeRF360 | | | | | Deep Blending | | | | | Tanks&Temples | | | | |
|---|---|---|---|---|---|---|---|---|---|---|---|---|---|---|---|
| | SSIM↑ | PSNR↑ | LPIPS↓ | GS Num↓ | Train↓ | SSIM↑ | PSNR↑ | LPIPS↓ | GS Num↓ | Train↓ | SSIM↑ | PSNR↑ | LPIPS↓ | GS Num↓ | Train↓ |
| Plenoxels* | 0.626 | 23.08 | 0.463 | - | 25m49s | 0.719 | 23.06 | 0.510 | - | 27m49s | 0.719 | 21.08 | 0.379 | - | 25m5s |
| INGP-Big* | 0.699 | 25.59 | 0.331 | - | 7m30s | 0.817 | 24.96 | 0.390 | - | 8m | 0.745 | 21.92 | 0.305 | - | 6m59s |
| M-NeRF360* | 0.792 | 27.69 | 0.237 | - | 48h | 0.901 | 29.40 | 0.245 | - | 48h | 0.759 | 22.22 | 0.257 | - | 48h |
| 3DGS(Kerbl et al., 2023) | 0.813 | 27.48 | 0.219 | 3337659 | 25m54s | 0.904 | 29.57 | 0.244 | 2832494 | 24m15s | 0.847 | 23.68 | 0.177 | 1847041 | 11m39s |
| AbsGS(Ye et al., 2024) | 0.795 | 27.01 | 0.217 | 3515320 | 25m25s | 0.901 | 29.60 | 0.237 | 1934586 | 17m56s | 0.852 | 23.65 | 0.168 | 1319894 | 10m26s |
| Pixel-GS(Zhang et al., 2024) | 0.821 | 27.66 | 0.200 | 5249325 | 30m17s | 0.903 | 29.87 | 0.232 | 2538913 | 21m51s | 0.854 | 23.89 | 0.161 | 2928914 | 15m21s |
| 3DGS with Ours | 0.825 | 27.86 | 0.201 | 2149503 | 19m57s | 0.908 | 30.07 | 0.230 | 1660517 | 17m49s | 0.856 | 24.15 | 0.167 | 1098518 | 9m56s |
| AbsGS with Ours | 0.823 | 27.73 | 0.199 | 2212337 | 19m41s | 0.906 | 30.09 | 0.224 | 1498901 | 14m12s | 0.857 | 23.98 | 0.155 | 947082 | 8m19s |

Table 1: **Quantitative comparison on Mip-NeRF 360, Deep Blending, and Tanks and Temples.** Metrics include SSIM↑, PSNR↑, LPIPS↓, Gaussian count (GS Num↓), and training time (Train↓). Cells are color-coded as best , second-best , and third-best . "with Ours" indicates the integration of our robust initialization and pruning strategy into base pipelines (3DGS or AbsGS). Asterisks (*) denote results reproduced from the original 3DGS paper: Plenoxels (Yu et al., 2021), INGP-Big (Müller et al., 2022), and Mip-NeRF360 (Barron et al., 2022). All other results are obtained under a unified training environment.

at approximately 4185px resolution, featuring unbounded 360-degree camera trajectories. **Tanks and Temples** (Knapitsch et al., 2017) provides large-scale outdoor scenes with complex geometric structures; we select the Train and Truck scenes. **Deep Blending** (Hedman et al., 2018) offers indoor environments with intricate details; we utilize the DrJohnson and Playroom scenes.

**Compared Methods.** We compare our method against representative methods across different scene representation paradigms: **(1) Non-Gaussian Methods: Mip-NeRF 360** (Barron et al., 2022) leverages cone tracing and multiscale encoding for unbounded outdoor reconstruction. **INGP-Big** (Müller et al., 2022) improves NeRF efficiency with hash-based feature grids. **Plenoxels** (Yu et al., 2021) represents radiance fields using sparse voxel grids with learned spherical harmonic coefficients. **(2) Gaussian Splatting Methods: 3DGS** (Kerbl et al., 2023) models the scene as a collection of 3D Gaussians rendered with differentiable splatting. **AbsGS** (Ye et al., 2024) enhances geometry learning using absolute image gradients during Gaussian densification. **Pixel-GS** (Zhang et al., 2024) improves the growth condition by considering pixel coverage weights to better handle large Gaussians and reduce floaters. We also include recent methods focusing on compression and density control: **MaskGaussian** (Liu et al., 2025) for adaptive pruning, **Perceptual-GS** (Zhou & Ni, 2025) for perceptual densification, and **SteepGS** (Wang et al., 2025b) for steepest descent density control. Detailed per-scene comparisons with the recent methods are provided in the appendix B.4.

**Implementation Details.** Following the standard 3DGS evaluation protocol, every eighth frame is reserved for testing. All experiments are conducted on an NVIDIA A800 GPU using the open-source 3DGS framework[1].

Each scene is trained for 30,000 iterations to ensure convergence. To maintain training stability, both dual-stage pruning and densification are scheduled within the first 9,000 iterations. Specifically, we modify two key hyperparameters from the original 3DGS setup to better suit our pipeline:

(1) The densification interval is increased from 100 to 300 iterations, which allows existing Gaussians to optimize more stably before introducing new ones. (2) The splitting gradient threshold is reduced from $2 \times 10^{-4}$ to $1 \times 10^{-4}$, encouraging finer Gaussian generation and improving detail reconstruction.

For KL-based Rendering Survival Pruning (KL-RSP), candidate Gaussians are selected if their gradient magnitude falls below $3 \times 10^{-6}$ and their spatial scale is below $5 \times 10^{-4}$ times the scene extent. The KL divergence threshold starts at $1 \times 10^4$ and gradually increases during training to progressively tighten pruning.

For Opacity-based Density-Constrained Pruning (ODCP), the initial opacity threshold is set to 0.1 and decreases over training to preserve low-opacity details. The opacity reset threshold is dynamically updated with a decay coefficient applied to the current value to maintain stability.

---

[1]3DGS for Real-Time Radiance Field Rendering: `https://github.com/graphdeco-inria/gaussian-splatting`

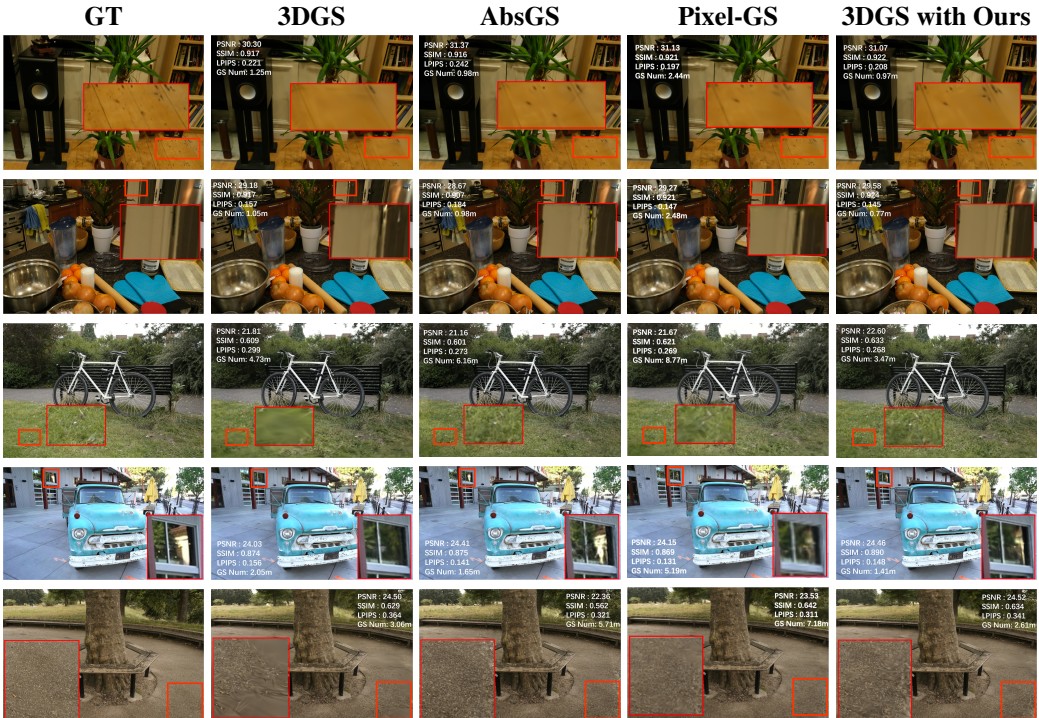

Figure 3: **Qualitative comparisons on representative scenes.** Visual results on Mip-NeRF 360 (room, counter, bicycles, treehill) and Tanks&Temples (truck) show that GRASP-GS improves rendering quality across both 3DGS and AbsGS pipelines. Compared to the baselines, it preserves more fine details, recovers sharper edges and textures, and effectively suppresses boundary blur and structural noise. Notably, our method handles reflective and specular regions more accurately. Each image includes per-scene PSNR, SSIM, LPIPS, and Number of Gaussians to validate improvements in both fidelity and compactness. Extended comparisons with GRASP integrated into AbsGS are provided in Figure 8 of Appendix B.6.

Performance is evaluated using standard metrics: PSNR, SSIM, and LPIPS[2] for image quality, along with the final number of retained Gaussians and total training time for efficiency analysis.

## 4.2 EXPERIMENTAL RESULTS

Table 1 summarizes quantitative results across three diverse datasets, with our method integrated into 3DGS and AbsGS pipelines denoted as "3DGS with Ours" and "AbsGS with Ours". Both variants consistently surpass their corresponding baselines across most metrics, highlighting the effectiveness of our method.

Compared to vanilla 3DGS, our method improves perceptual quality (LPIPS reduced from 0.219 to 0.201 on Mip-NeRF360) while reducing Gaussian count by 35.6% (from 3.3M to 2.1M) and training time by 23% (from 25min54s to 19min57s). Our approach also demonstrates competitive performance against other state-of-the-art methods while requiring substantially less computational resources. The initialization overhead introduced by our geometric prior-guided module is as detailed in Appendix B.3. Comprehensive efficiency analysis, including GPU memory usage, rendering FPS comparison, pruning overhead, and training progression, is provided in Appendix B.2.

**Qualitative comparison.** Figure 3 demonstrates GRASP-GS's improved visual quality compared to vanilla 3DGS and recent Gaussian splatting methods. Our approach consistently enhances rendering fidelity through better geometric priors and adaptive pruning.

---

[2]PSNR measures peak signal-to-noise ratio (higher is better), SSIM evaluates structural similarity (higher is better), and LPIPS assesses learned perceptual image patch similarity (lower is better).

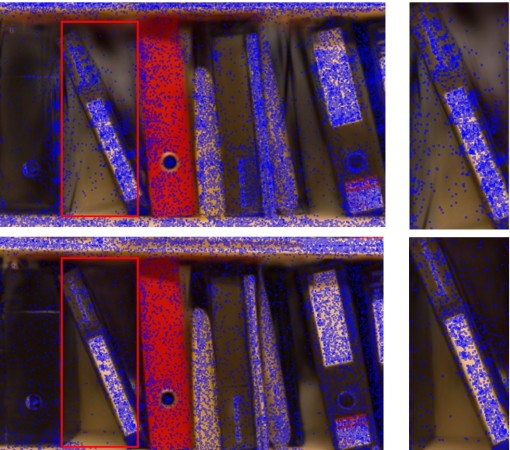

Figure 4: **Optimized Gaussian Structure by GRASP-GS.** Our method yields compact, semantically structured Gaussians that enhance geometry clarity and rendering quality.

As shown in Figure 3, GRASP-GS achieves sharper details and more accurate reflections in challenging scenarios like reflective surfaces (e.g., truck windows) and texture-rich regions (e.g., room details). The visual improvements are accompanied by metric gains in PSNR, SSIM, and LPIPS while reducing Gaussian counts.

To demonstrate the general applicability of our modules, we also integrate GRASP into the AbsGS framework. As shown in Figure 8 of Appendix B.6, our method brings consistent improvements to AbsGS, confirming the transferability of our technical contributions.

**Spatial optimization analysis.** Figure 4 visualizes the distribution of Gaussian centers, showing that GRASP-GS produces a more compact, semantically structured layout compared to baseline methods. This regularized distribution avoids unnecessary overlap and enhances geometric clarity, contributing to lower memory and computation costs while achieving high visual fidelity. Comprehensive depth map comparisons between our method and baseline 3DGS across all Mip-NeRF360 scenes are provided in Appendix B.1, demonstrating superior geometric accuracy achieved through the combined effect of our geometric prior-guided initialization and dual-stage pruning strategy.

## 4.3 ABLATION STUDY

We conduct ablation experiments to evaluate the contributions of our two core modules in GRASP-GS: Geometric Registration (Reg.) and Adaptive Saliency Pruning (Prune).

**Core Module Ablation.** Table 2 presents the impact of each module individually and in combination. Geometric Registration alone improves rendering quality with +0.29 dB PSNR and +0.009 SSIM gains by refining point cloud initialization through geometric priors, effectively reducing artifacts inherent in sparse SfM reconstruction. This improvement comes with modest computational overhead, including 9% more Gaussians and 11% longer training time due to denser initialization.

| Reg. | Prune | PSNR↑ | SSIM↑ | LPIPS↓ | GS Num↓ | Time↓ |
|------|-------|-------|-------|--------|---------|-------|
| $X$ | $X$ | 27.48 | 0.813 | 0.219 | 3.3M | 25m54s |
| $\checkmark$ | $X$ | 27.77 | 0.822 | **0.199** | 3.7M | 28m51s |
| $X$ | $\checkmark$ | 27.63 | 0.818 | 0.212 | **2.0M** | **19m33s** |
| $\checkmark$ | $\checkmark$ | **27.86** | **0.825** | 0.201 | 2.1M | 19m57s |

Table 2: Ablation on core modules: Geometric Registration (Reg.) and Adaptive Saliency Pruning (Prune). Reg. improves rendering quality, while Prune enhances efficiency.

Adaptive Saliency Pruning demonstrates strong efficiency, reducing the Gaussian count by 41% (from 3.34M to 1.99M) and training time by 25% while maintaining competitive quality (+0.15 dB PSNR). When both modules are combined, the framework achieves the best balance, with +0.38 dB PSNR, +0.012 SSIM improvements, and 0.018 LPIPS reduction compared to baseline. The integrated approach also maintains substantial gains, with 36% fewer Gaussians and 23% faster training. This synergy stems from the complementary feedback loop: initialization alone achieves +0.29 dB PSNR but with 3.7M Gaussians (28m51s), while pruning alone reduces to 2.0M Gaussians (19m33s) but only 27.63 dB PSNR. The combination achieves 27.86 dB PSNR with 2.1M Gaussians (19m57s), demonstrating that better initialization allows pruning to focus on genuine redundancies, while pruning preserves and refines the geometric structure from initialization rather than degrading it.

**Impact of Prior Quality.** Table 3 evaluates the influence of prior quality and architectural choices on performance. To verify whether gains stem solely from the quality of the MoGe prior, we conduct several complementary experiments on Mip-NeRF360.

- **Alternative Prior**: We replace MoGe with the weaker Depth Anything prior while keeping the full GRASP-GS pipeline. Despite the noisier input, the method still achieves competitive performance, indicating robustness to prior quality. We also evaluate advanced geometric foundation models (VGGT) (Wang et al., 2025a) as priors; detailed results are provided in Appendix C3.

- **Bypass Ablation**: We use only the registration module without the geometric-chromatic consistency filtering, while keeping the dual-stage pruning strategy. The resulting performance drops notably, showing that GRASP-GS is essential for distilling geometric-chromatic consistency rather than merely relying on good priors.

- **Multi-View Comparison**: Our MoGe-based approach achieves superior performance over MASt3R (+2.24 dB PSNR on average), demonstrating that point cloud quality and our filtering strategy are more critical than multi-view consistency alone when proper geometric registration is performed (detailed results in Appendix C4).

| Method | SSIM↑ | PSNR↑ | LPIPS↓ | GS Num↓ | Time↓ |
|---|---|---|---|---|---|
| Ours (AP) | 0.823 | 27.81 | 0.203 | 2.2M | 20m02s |
| MoGe (BA) | 0.813 | 27.35 | 0.203 | 2.7M | 22m13s |
| MASt3R (MV) | 0.780 | 25.62 | 0.234 | 2.6M | 21m07s |
| Ours | **0.825** | **27.86** | **0.201** | **2.1M** | **19m57s** |

Table 3: Impact of prior quality and architectural ablation. "Ours (AP)" uses a weaker prior (Depth Anything); "MoGe (BA)" uses only the registration module, while keeping dual-stage pruning; "MASt3R (MV)" uses multi-view dense priors with GRASP-GS modules. Results show our method is robust to prior quality and not merely a wrapper.

## 5 CONCLUSION

We present GRASP-GS, a modular yet integrated enhancement for 3DGS that combines geometric prior-guided initialization with dual-stage saliency pruning. Dense geometric priors and consistency filtering produce high-quality initial Gaussians, while KL-based and opacity-based pruning reduce redundancy and preserve structural fidelity.

Our approach reduces Gaussian count by 35.6% and training time by 23% compared to vanilla 3DGS, while improving PSNR, SSIM, and LPIPS across multiple benchmarks, demonstrating efficient and high-fidelity rendering. Importantly, GRASP-GS is flexible and generalizable, allowing straightforward integration into other 3D Gaussian-based variants or extended frameworks to further improve initialization, pruning, and scene representation quality.

Future work includes extending GRASP-GS to dynamic and large-scale scenes, incorporating learned or scene-adaptive geometric priors, and developing hierarchical or multi-resolution pruning strategies. Additionally, exploring cross-modal priors and semantic-guided Gaussian optimization could further enhance scalability, robustness, and fidelity in complex environments.

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

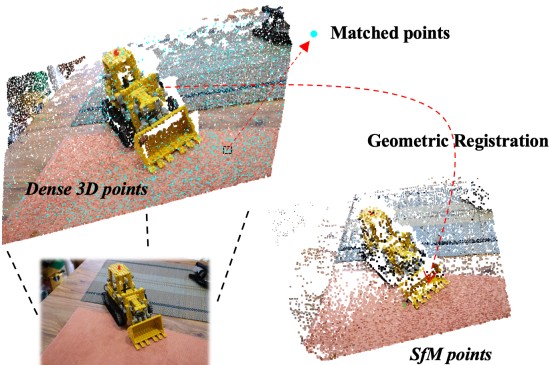

Figure 5: Dense 3D points are predicted by MoGe from single-view images. Points that successfully match sparse SfM points based on 2D pixel correspondences are highlighted in cyan as matched points. These matched points serve as reliable 3D-3D anchors for evaluating geometric and chromatic consistency during viewpoint filtering.

## A  SUPPLEMENTARY METHODOLOGY

### A.1  GEOMETRIC REGISTRATION DETAILS

To achieve accurate alignment between dense point clouds predicted by MoGe and the global sparse SfM reconstruction(see Figure 5), we adopt a coarse-to-fine geometric registration strategy:

- **Initial Alignment via RANSAC.** A robust transformation is estimated using 3D-3D correspondences under the RANSAC (Fischler & Bolles, 1981) framework, which rejects outliers while maximizing inlier support.

- **Similarity Transformation via Kabsch-Umeyama.** We compute a similarity transform—including rotation, translation, and uniform scaling—using the Kabsch-Umeyama algorithm (Lawrence et al., 2019), providing a closed-form optimal alignment under least-squares error.

- **Refinement via ICP.** Finally, alignment is refined using Iterative Closest Point (ICP) (Besl & McKay, 1992), minimizing point-to-point distances between overlapping regions to produce tightly registered geometry.

This hierarchical strategy ensures both robustness and precision, enabling reliable consistency loss computation in the main pipeline.

### A.2  CANDIDATE SELECTION VIA SPATIAL SIMILARITY

A subset of pruning candidates $\mathcal{G}_{\text{cand}}$ is selected by filtering Gaussians with low optimization gradients and large spatial extent:

$$\mathcal{G}_{\text{cand}} = \{G_i \mid \|\nabla G_i\| < \tau_{\text{grad}}, \ \lambda_{\max}(S_i) > \tau_{\text{scale}}\} \tag{8}$$

Here, $\tau_{\text{grad}}$ excludes Gaussians in unstable regions that are still being actively optimized, while $\tau_{\text{scale}}$ preserves fine-scale structures essential for detailed geometry.

To evaluate local similarity, the 3D space is partitioned into uniform voxels with edge length $v_{\text{size}}$. Each Gaussian $G_i$ is assigned to a voxel based on its spatial mean $\mu_i$:

$$\mathbf{v}_i = \left\lfloor \frac{\mu_i}{v_{\text{size}}} \right\rfloor \tag{9}$$

To preserve scene structure, only voxels containing sufficient Gaussians are retained. The minimum number of Gaussians per voxel is defined as:

$$N_{\min}^{\text{voxel}} = \max\left( \frac{|\mathbf{G}|}{\kappa}, \ 2 \right) \tag{10}$$

where $|\mathbf{G}|$ denotes the total number of Gaussians and $\kappa$ is a constant determined empirically to balance memory usage and spatial filtering robustness.

Within each retained voxel, pairwise KL divergence is computed for all Gaussian pairs $G_i(\mu_i, \Sigma_i)$ and $G_j(\mu_j, \Sigma_j)$ using:

$$D_{\mathrm{KL}}(G_i \parallel G_j) = \frac{1}{2}\Big[ \mathrm{tr}(\Sigma_j^{-1}\Sigma_i) + (\mu_j - \mu_i)^\top \Sigma_j^{-1}(\mu_j - \mu_i)$$
$$- \log \frac{|\Sigma_i|}{|\Sigma_j|} - d \Big] \tag{11}$$

where $d = 3$ is the spatial dimension. This metric captures similarity in position, orientation, and scale, offering a more comprehensive measure than Euclidean distance.

A sparse similarity matrix $\mathbf{S}$ is constructed by retaining only the lower-triangular elements with $S_{ij} < \tau_{\mathrm{KL}}$ to reduce the number of Gaussian pairs considered. The resulting qualified Gaussian pairs are denoted as:

$$\mathbf{P} = \{(G_i, G_j) \mid i > j,\ S_{ij} < \tau_{\mathrm{KL}}\} \tag{12}$$

## B  Supplementary Experiments

### B.1  Depth Map Visualization and Geometric Analysis

To validate the geometric improvements achieved by our geometric prior-guided initialization, we provide comprehensive depth map comparisons between our method and the baseline 3DGS across some scenes in the Mip-NeRF360 dataset. Depth visualization provides a direct and quantitative way to evaluate geometric accuracy, which is particularly important for validating that our MoGe-based initialization leads to improved geometry.

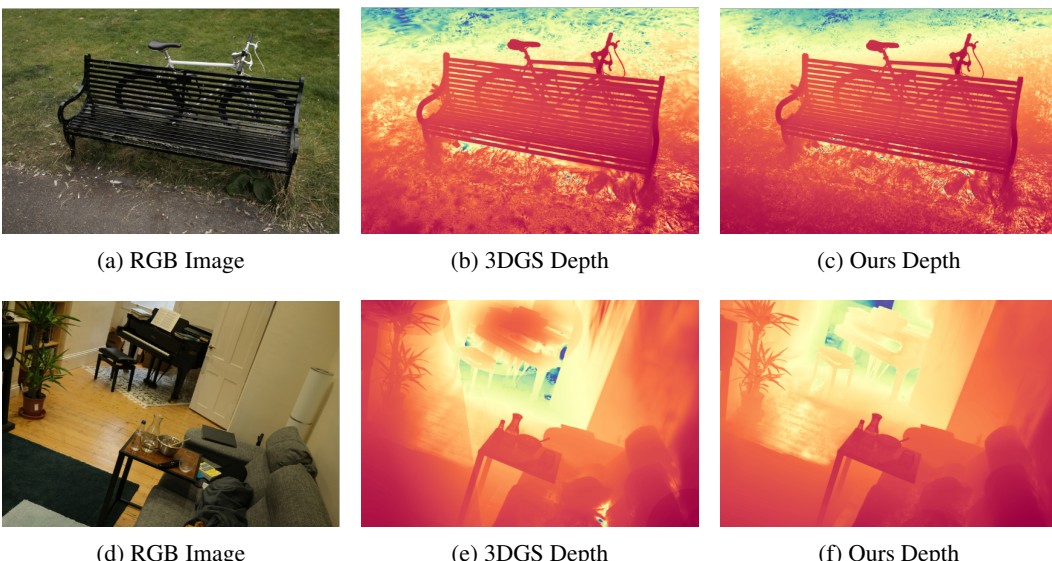

|                 |                 |                 |
|:---------------:|:---------------:|:---------------:|
| (a) RGB Image   | (b) 3DGS Depth  | (c) Ours Depth  |
| (d) RGB Image   | (e) 3DGS Depth  | (f) Ours Depth  |

Figure 6: Depth map comparison for representative scenes. Top row: Bicycle scene. Bottom row: Room scene. Our method demonstrates superior geometric accuracy with more precise depth boundaries and better preservation of fine-scale structures compared to the baseline 3DGS.

The depth map comparisons clearly show that our geometric prior-guided initialization and dual-stage pruning strategy work synergistically to transfer geometric knowledge from foundation models to the final Gaussian representation, resulting in more accurate geometry despite using fewer Gaussians.

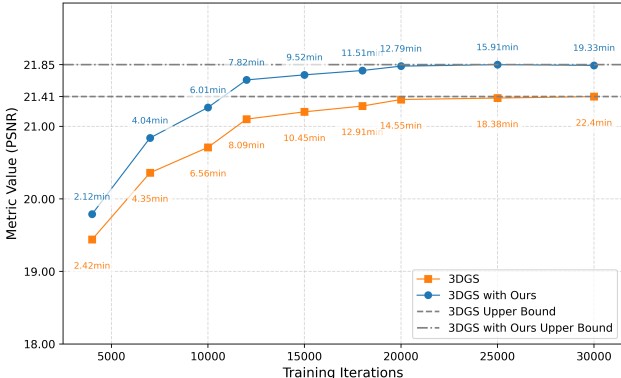

Figure 7: **Training progression on the flowers scene (based on 3DGS).** We compare the training efficiency between 3DGS and our GRASP-GS variant built upon 3DGS. PSNR is plotted over iterations, with time annotations. GRASP-GS achieves higher rendering quality with fewer iterations and reduced training time.

## B.2 EFFICIENCY ANALYSIS

Comprehensive profiling is conducted to measure GPU memory usage during training, rendering FPS, and pruning operation overhead. Table 4 presents detailed results across all Mip-NeRF360 scenes.

| Scene | Peak Memory (GB) | 3DGS FPS | Ours FPS | FPS ↑ | Prune Time (s) | Prune Ops | Prune % |
|-------|------------------|----------|----------|-------|----------------|-----------|---------|
| Bicycle | 31.06 | 94.91 | 153.12 | +61.3% | 8.35 | 9 | 0.79% |
| Kitchen | 22.31 | 166.15 | 239.24 | +44.0% | 6.76 | 9 | 0.68% |
| Room | 39.84 | 182.22 | 239.19 | +31.2% | 6.51 | 9 | 0.67% |
| Counter | 40.31 | 197.18 | 247.60 | +25.6% | 7.41 | 9 | 0.78% |
| Flowers | 14.90 | 204.31 | 214.78 | +5.1% | 4.67 | 9 | 0.48% |
| Stump | 12.73 | 164.16 | 192.36 | +17.2% | 4.30 | 9 | 0.48% |
| Treehill | 12.88 | 173.09 | 194.68 | +12.5% | 4.58 | 9 | 0.48% |
| Bonsai | 21.08 | 264.06 | 276.79 | +4.8% | 4.65 | 9 | 0.53% |
| Garden | 18.63 | 131.71 | 166.41 | +26.3% | 4.21 | 9 | 0.39% |
| **Average** | **23.75** | **175.31** | **213.80** | **+22.0%** | **5.72** | **9** | **0.59%** |

Table 4: Comprehensive efficiency profiling across all Mip-NeRF360 scenes, including GPU memory usage, rendering FPS comparison with baseline 3DGS, and pruning operation overhead. Peak memory occurs during pruning operations due to KL-divergence computations for large Gaussian sets, which can be reduced through batch processing at the cost of increased pruning time. Our method achieves an average FPS improvement of +22.0% (213.80 vs 175.31 FPS) with minimal pruning overhead (0.59% of training time).

The profiling results demonstrate that our method achieves significant efficiency improvements: (1) Rendering Performance: Average FPS improvement of +22.0% (213.80 vs 175.31 FPS for baseline 3DGS). FPS improvements directly correlate with Gaussian count reduction, validating our compact representation claims. (2)Pruning Overhead: The dual-stage pruning strategy incurs minimal overhead (average 5.72 seconds, 0.59% of training time), with operations strategically spaced at 1000-iteration intervals. (3)Memory Usage: Peak GPU memory usage averages 23.75 GB (range: 12.7-40.3 GB), primarily due to KL-divergence computations during pruning operations involving large Gaussian sets. Memory usage can be reduced through batch processing, but this would increase pruning time. For training efficiency analysis, we select the flowers scene from the Mip-NeRF 360 dataset and compare the training progression of GRASP-GS against the original 3DGS. As shown in Figure 7, our method achieves higher PSNR with fewer iterations and significantly reduced training time. GRASP-GS reaches high-quality results early in training, making it well-suited for applications demanding rapid convergence or real-time adaptability. This demonstrates the robustness and temporal efficiency of our optimization strategy under constrained training budgets.

### B.3 INITIALIZATION OVERHEAD ANALYSIS

To provide a complete efficiency evaluation, we report the computational overhead introduced by our geometric prior-guided initialization module. The initialization consists of two stages: (1) MoGe depth estimation for selected views, and (2) point cloud alignment and fusion with sparse SfM points.

**Time Breakdown:**

1. **MoGe Inference**: We use batch inference mode, processing all 24 selected views in a single forward pass. The core inference time is 4.23 seconds. Additional I/O overhead for the operational pipeline includes image loading ( 5.8 seconds), excluding optional debug exports (e.g., dense EXR dumping) that can be removed without affecting the method.

2. **Point Cloud Fusion**: The fusion process is embarrassingly parallel over the 24 selected views. The total core computation corresponding to aligning and fusing all views is approximately 2.5 seconds, which can be realized in practice via view-wise parallel execution. Additional I/O overhead includes loading COLMAP sparse points ( 3.9 seconds) and saving the final fused point cloud.

**Total Initialization Overhead:** The total initialization overhead is **6.7 seconds** (4.23s MoGe inference + 2.5s point cloud fusion) for core computation. Including I/O operations (image loading, COLMAP point loading, and final point cloud saving), the total overhead is approximately 17 seconds per scene.

### B.4 COMPARISON WITH RECENT SOTA METHODS

Comprehensive comparisons are conducted with three recent state-of-the-art methods published in 2025 that represent different categories of 3DGS improvements: MaskGaussian (Liu et al., 2025) for compression/pruning, Perceptual-GS (Zhou & Ni, 2025) for density control, and SteepGS (Wang et al., 2025b) for theoretically-grounded density control. Tables 5, 6, and 7 present detailed per-scene quantitative results comparing these methods with our approach across all Mip-NeRF360 scenes, providing insights into the behavior of our method under different geometric structures and content complexities.

| Scene | Ours | | | Ours | | MaskGaussian | | | MaskGaussian | |
|---|---|---|---|---|---|---|---|---|---|---|
| | PSNR↑ | SSIM↑ | LPIPS↓ | GS Num↓ | Time↓ | PSNR↑ | SSIM↑ | LPIPS↓ | GS Num↓ | Time↓ |
| Bicycle | **25.44** | **0.774** | **0.204** | 3.54M | 22m12s | 25.12 | 0.748 | 0.245 | **2.88M** | 22m09s |
| Bonsai | **32.68** | **0.950** | **0.172** | 0.91M | 16m45s | 32.12 | 0.944 | 0.184 | **0.46M** | 14m30s |
| Counter | **29.32** | **0.920** | **0.168** | 0.81M | 18m45s | 28.40 | 0.913 | 0.188 | **0.42M** | 16m21s |
| Flowers | **21.85** | **0.612** | **0.337** | 2.55M | 16m10s | 21.34 | 0.587 | 0.362 | **1.79M** | 17m43s |
| Garden | **27.55** | **0.866** | **0.110** | 3.52M | 23m16s | 26.67 | 0.854 | 0.125 | **2.85M** | 21m24s |
| Kitchen | **32.04** | **0.936** | **0.109** | 0.89M | 19m44s | 31.38 | 0.931 | 0.119 | **0.79M** | 18m33s |
| Room | **32.32** | **0.936** | **0.174** | 0.86M | 19m44s | 31.53 | 0.925 | 0.201 | **0.37M** | 16m05s |
| Stump | **27.05** | **0.791** | **0.210** | 3.50M | 19m12s | 26.57 | 0.767 | 0.244 | **2.05M** | 17m28s |
| Treehill | **22.55** | **0.641** | **0.329** | 2.74M | 19m26s | 22.58 | 0.636 | 0.350 | **1.94M** | 17m30s |
| **Average** | **27.86** | **0.825** | **0.201** | 2.15M | 19m57s | 27.44 | 0.811 | 0.224 | **1.51M** | 18m00s |

Table 5: Per-scene comparison with MaskGaussian across all Mip-NeRF360 scenes. Our method achieves better reconstruction quality (average improvement: +0.42 dB PSNR, +0.014 SSIM, -0.023 LPIPS) while MaskGaussian achieves better compression (fewer Gaussians).

The comprehensive experimental comparison demonstrates that our method achieves superior or competitive performance compared to recent state-of-the-art methods across different categories of 3DGS improvements.

Compared to MaskGaussian (compression/pruning method), our approach achieves better reconstruction quality (average improvement: +0.42 dB PSNR, +0.014 SSIM, -0.023 LPIPS) as shown in Table 5. While MaskGaussian achieves better compression with fewer Gaussians (1.51M vs 2.15M on average), our method provides superior quality-efficiency trade-off by combining high-quality initialization with effective pruning.

| Scene | Ours | | | Ours | | Perceptual-GS | | | Perceptual-GS | |
|---|---|---|---|---|---|---|---|---|---|---|
| | PSNR↑ | SSIM↑ | LPIPS↓ | GS Num↓ | Time↓ | PSNR↑ | SSIM↑ | LPIPS↓ | GS Num↓ | Time↓ |
| Bicycle | 25.44 | 0.774 | 0.204 | **3.54M** | **22m12s** | **25.97** | **0.804** | **0.166** | 3.90M | 27m14s |
| Bonsai | 32.68 | 0.950 | 0.172 | **0.91M** | **16m45s** | **32.69** | **0.952** | **0.152** | 1.56M | 28m39s |
| Counter | 29.32 | 0.920 | 0.168 | **0.81M** | **18m45s** | **29.37** | **0.921** | **0.157** | 1.52M | 35m36s |
| Flowers | **21.85** | 0.612 | 0.337 | **2.55M** | **16m10s** | 21.82 | **0.654** | **0.257** | 3.53M | 27m29s |
| Garden | 27.55 | 0.866 | 0.110 | 3.52M | 23m16 | **27.91** | **0.876** | **0.099** | **3.04M** | 23m59s |
| Kitchen | **32.04** | **0.936** | 0.109 | **0.89M** | **19m44s** | 31.63 | 0.935 | **0.107** | 1.64M | 35m38s |
| Room | **32.32** | **0.936** | 0.174 | **0.86M** | **19m44s** | 32.11 | 0.935 | **0.170** | 1.76M | 34m26s |
| Stump | 27.05 | 0.791 | 0.210 | **3.50M** | **19m12s** | **27.26** | **0.805** | **0.177** | 3.80M | 25m44s |
| Treehill | **22.55** | 0.641 | 0.329 | **2.74M** | **19m26s** | 22.54 | **0.656** | **0.274** | 3.51M | 26m54s |
| **Average** | 27.86 | 0.825 | 0.201 | **2.15M** | **19m57s** | 27.92 | 0.838 | 0.173 | 2.70M | 29m30s |

Table 6: Per-scene comparison with Perceptual-GS across all Mip-NeRF360 scenes. Perceptual-GS achieves slightly better average quality metrics, while our method achieves better efficiency (fewer Gaussians: 2.15M vs 2.70M, faster training: 19m57s vs 29m30s).

| Scene | Ours | | | Ours | | SteepGS | | | SteepGS | |
|---|---|---|---|---|---|---|---|---|---|---|
| | PSNR↑ | SSIM↑ | LPIPS↓ | GS Num↓ | Time↓ | PSNR↑ | SSIM↑ | LPIPS↓ | GS Num↓ | Time↓ |
| Bicycle | **25.44** | **0.774** | **0.204** | **3.54M** | **22m12s** | 25.12 | 0.747 | 0.244 | 5.75M | 33m13s |
| Bonsai | **32.68** | **0.950** | **0.172** | **0.91M** | 16m45s | 32.25 | 0.946 | 0.180 | 1.24M | 20m01s |
| Counter | **29.32** | **0.920** | **0.168** | **0.81M** | 18m45s | 29.05 | 0.914 | 0.184 | 1.16M | 21m56s |
| Flowers | **21.85** | **0.612** | **0.337** | **2.55M** | **16m10s** | 21.38 | 0.588 | 0.360 | 3.40M | 25m35s |
| Garden | **27.55** | **0.866** | **0.110** | **3.52M** | **23m16s** | 27.32 | 0.856 | 0.122 | 5.67M | 33m51s |
| Kitchen | **32.04** | **0.936** | **0.109** | **0.89M** | **19m44s** | 31.54 | 0.932 | 0.117 | 1.74M | 25m34s |
| Room | **32.32** | **0.936** | **0.174** | **0.86M** | **19m44s** | 31.60 | 0.926 | 0.197 | 1.48M | 22m49s |
| Stump | **27.05** | **0.791** | **0.210** | **3.50M** | **19m12s** | 26.70 | 0.770 | 0.242 | 4.44M | 26m56s |
| Treehill | 22.55 | **0.641** | **0.329** | **2.74M** | **19m26s** | 22.64 | 0.636 | 0.347 | 3.44M | 24m59s |
| **Average** | 27.86 | 0.825 | 0.201 | **2.15M** | **19m57s** | 27.51 | 0.813 | 0.221 | 3.15M | 26m11s |

Table 7: Per-scene comparison with SteepGS across all Mip-NeRF360 scenes. Our method achieves better quality-efficiency trade-off (average improvement: +0.35 dB PSNR, +0.012 SSIM, -0.020 LPIPS, with fewer Gaussians: 2.15M vs 3.15M).

Compared to Perceptual-GS (density control method), our method achieves comparable quality with better efficiency as shown in Table 6. While Perceptual-GS achieves slightly better average quality metrics (27.92 vs 27.86 dB PSNR, 0.838 vs 0.825 SSIM, 0.173 vs 0.201 LPIPS), our method achieves better efficiency (fewer Gaussians: 2.15M vs 2.70M, faster training: 19m57s vs 29m30s).

Compared to SteepGS (theoretically-grounded density control), our method achieves better quality-efficiency trade-off as shown in Table 7 (average improvement: +0.35 dB PSNR, +0.012 SSIM, -0.020 LPIPS, with fewer Gaussians: 2.15M vs 3.15M).

## B.5 DETAILED PER-SCENE ANALYSIS

Table 8 provides a comprehensive breakdown of per-scene performance for GRASP-GS built upon the 3DGS framework, reporting PSNR, SSIM, LPIPS, training time, and the final number of retained Gaussians for each scene across three datasets. This detailed analysis offers insights into the behavior of our method under different geometric structures and content complexities.

| Dataset Metric | Mip-NeRF360 | | | | | | | | | Tanks&Temples | | Deep Blending | |
|---|---|---|---|---|---|---|---|---|---|---|---|---|---|
| | *flowers* | *treehill* | *garden* | *bicycle* | *kitchen* | *stump* | *room* | *counter* | *bonsai* | *truck* | *train* | *Drjohnson* | *Playroom* |
| **PSNR** ↑ | 21.85 | 22.55 | 27.55 | 25.44 | 32.04 | 27.05 | 32.32 | 29.32 | 32.68 | 25.82 | 22.49 | 29.68 | 30.47 |
| **SSIM** ↑ | 0.612 | 0.641 | 0.866 | 0.774 | 0.936 | 0.791 | 0.935 | 0.920 | 0.950 | 0.888 | 0.826 | 0.906 | 0.911 |
| **LPIPS** ↓ | 0.337 | 0.329 | 0.110 | 0.203 | 0.109 | 0.210 | 0.174 | 0.168 | 0.172 | 0.139 | 0.197 | 0.230 | 0.231 |
| **Train Time** (hh:mm) | 19:10 | 19:26 | 23:16 | 22:12 | 20:14 | 19:12 | 19:44 | 18:45 | 16:45 | 8:20 | 8:22 | 17:10 | 15:12 |
| **GS Num** (Millions) | 2.55 | 2.74 | 3.52 | 3.54 | 0.89 | 3.50 | 0.86 | 0.81 | 0.91 | 1.35 | 0.71 | 1.79 | 1.23 |

Table 8: Per-scene performance on three datasets. Metrics include PSNR ↑, SSIM ↑, LPIPS ↓, Training Time, and GS Num. These results reflect the efficiency and adaptability of GRASP-GS across diverse scene types.

The results reveal several key insights: (1) Indoor scenes like *kitchen*, *room*, and *counter* achieve consistently high SSIM values (> 0.90) with relatively compact Gaussian representations, indicating

effective geometric regularization in structured environments. (2) Outdoor scenes such as *flowers* and *treehill* require more Gaussians but maintain reasonable training times, demonstrating scalability across scene complexities. (3) Challenging scenarios including *garden* and *Drjohnson* maintain high rendering quality with efficient representations, confirming the robustness and adaptability of GRASP-GS in diverse geometric structures and lighting conditions.

## B.6 ABSGS WITH OURS

To demonstrate the general applicability of our geometric initialization and saliency pruning modules, we integrate GRASP into the AbsGS framework. As shown in Figure 8, GRASP brings consistent improvements to AbsGS across all test scenes, particularly in handling fine details and suppressing artifacts.

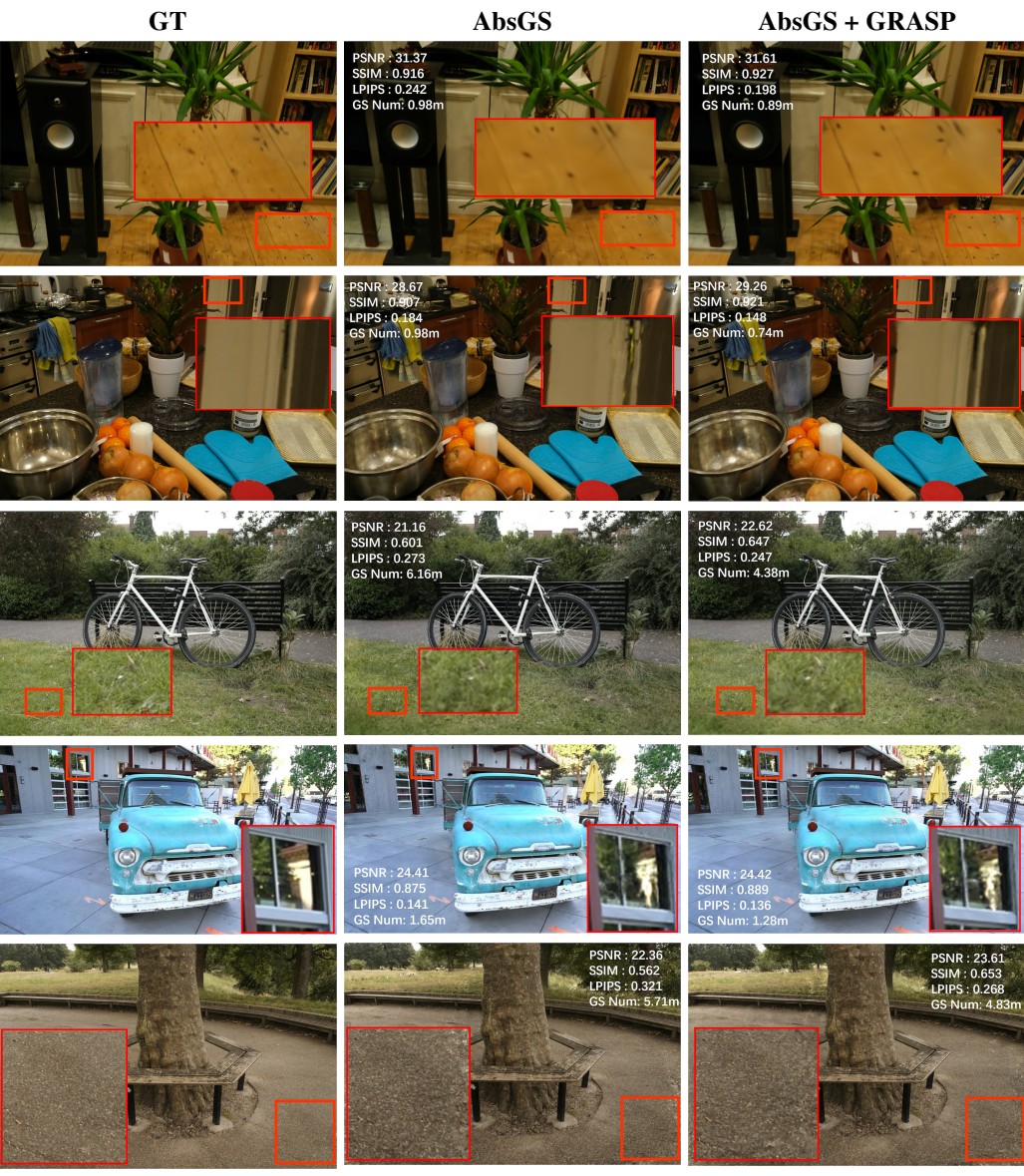

Figure 8: **GRASP integrated with AbsGS.** The geometric initialization and saliency pruning modules of GRASP can be effectively integrated into other Gaussian Splatting frameworks. Quantitative improvements are shown in Table 1, with consistent gains in PSNR, SSIM, and LPIPS while reducing Gaussian counts.

The integration results confirm that our technical contributions are modular and transferable. GRASP's components provide consistent benefits regardless of the underlying GS framework, demonstrating their general applicability in improving Gaussian Splatting methods. The improvements are most noticeable in regions with complex textures and reflective surfaces, where better initialization leads to more accurate Gaussian placement and pruning removes redundant elements more effectively.

## C  DETAILED ABLATION STUDY

### C.1  GEOMETRIC REGISTRATION MODULE ANALYSIS

**Dense vs. Sparse Initialization** The geometric registration module addresses the fundamental limitation of sparse SfM reconstruction by generating dense point clouds with superior geometric coverage. As demonstrated in Figure 9, dense initialization significantly improves reconstruction quality, particularly in boundary regions where SfM typically fails to establish reliable correspondences. The dense approach provides more complete scene coverage, reducing missing geometry and structural artifacts that are commonly observed with sparse SfM-based initialization.

**Initialized with Structure-from-Motion (SfM)**

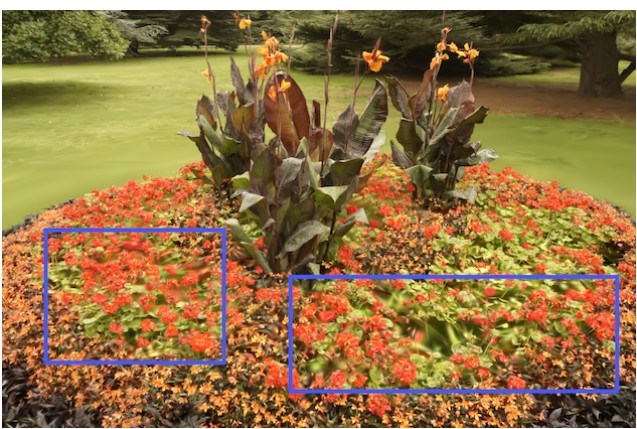

**Initialized with dense points**

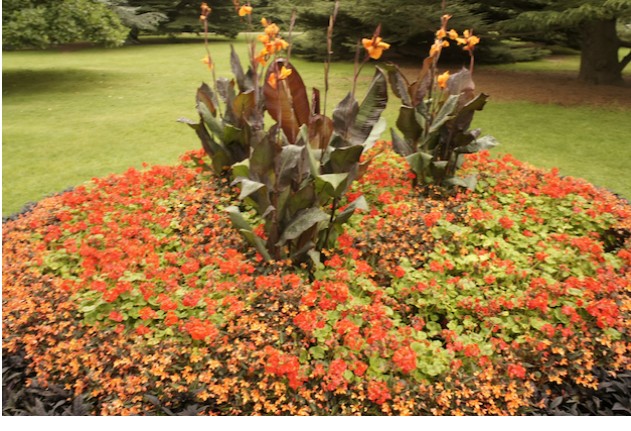

Figure 9: Qualitative comparison of 3D Gaussian initialization. Top: using standard SfM-based sparse points (baseline 3DGS without our initialization). Bottom: using dense point cloud initialization provided by our Geometric Prior-Guided module in GRASP-GS, showing improved geometric completeness and structural detail.

**Geometric-Chromatic Consistency Filtering** The filtering mechanism plays a crucial role in ensuring initialization quality by removing geometrically inconsistent points from dense point clouds.

Table 9 reveals a counterintuitive but important finding: without filtering, despite having 9% more initial points (2.57M vs 2.36M), the rendering quality deteriorates across all metrics. This demonstrates that accurate and consistent initialization is more critical than point density alone.

| Method | PSNR↑ | SSIM↑ | LPIPS↓ | GS Num↓ | Train Time↓ |
|---|---|---|---|---|---|
| w/o Filtering | 27.78 | 0.820 | 0.199 | 2,569,135 | 23m01s |
| Ours | 27.86 | 0.825 | 0.201 | 2,149,503 | 19m57s |

Table 9: Impact of geometric-chromatic consistency filtering. The filtering mechanism improves rendering quality while reducing computational overhead.

**Point Cloud Refinement** The refinement submodule specifically targets noise and artifacts inherent in monocular depth estimation techniques. Figure 10 clearly illustrates the effectiveness of this refinement process: the refined geometry (left) exhibits clean structural details with suppressed noise, while the unrefined geometry (right) contains residual artifacts and inconsistencies typical of raw monocular depth predictions.

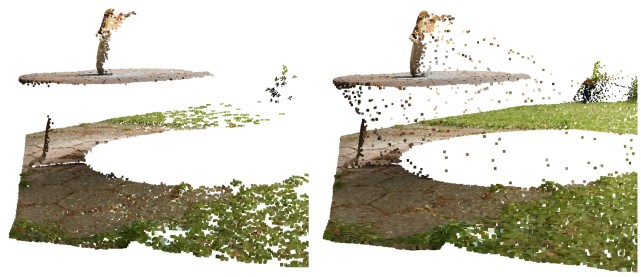

Figure 10: Effect of Point Cloud Refinement. Left: refined geometry with suppressed noise and eliminated trailing artifacts. Right: unrefined geometry showing artifacts from monocular depth estimates.

This refinement step is particularly crucial for maintaining geometric coherence in complex scenes where depth discontinuities and occlusions challenge monocular estimation accuracy. The clean geometric priors produced by refinement directly contribute to more stable and effective Gaussian initialization.

C.2    ADAPTIVE SALIENCY PRUNING MODULE ANALYSIS

**Two-Stage Pruning Effectiveness** The adaptive pruning strategy demonstrates remarkable effectiveness in removing redundant Gaussians while preserving essential scene representations. Figure 11 provides a compelling visual comparison: the pruned configuration (left) achieves significantly more compact Gaussian distributions with clean rendering quality, while the unpruned baseline (right) exhibits spatial redundancy and rendering artifacts due to excessive Gaussian density.

The two-stage approach targets different types of redundancy: spatial overlap in densely populated regions and temporal instability in inconsistently contributing Gaussians. This comprehensive strategy ensures that pruning maintains representational capacity while achieving substantial efficiency improvements.

**Opacity Reset Strategy Analysis** To stabilize pruning during training, an adaptive opacity reset mechanism is introduced. Instead of a fixed reset value, the reset threshold is initialized with $\tau_{opacity}$ and decays progressively. This gradual increase of prune-eligible Gaussians prevents abrupt opacity changes and avoids premature removal of informative components.

The opacity reset mechanism presents a complex trade-off that requires careful consideration. As shown in Table 10, while opacity resetting achieves a modest reduction in Gaussian count (1.96M vs 2.15M, approximately 9% reduction), it introduces several detrimental effects that compromise overall system performance.

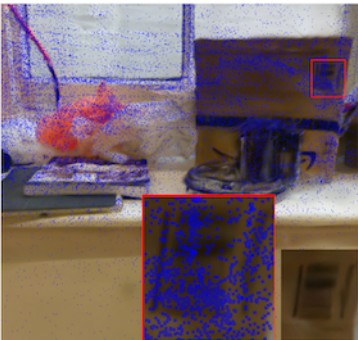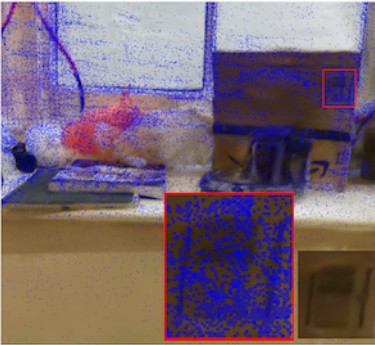

Figure 11: Effect of Adaptive Saliency Pruning. Left: compact pruned distribution with clean renderings. Right: redundant unpruned distribution exhibiting artifacts and inefficiencies.

| Method \ Metric | PSNR↑ | SSIM↑ | LPIPS↓ | GS Num↓ | Train Time↓ |
|---|---|---|---|---|---|
| **Raw Reset Opacity** | 27.74 | 0.8229 | 0.2071 | 1,962,881 | 21m31s |
| **Ours** | 27.86 | 0.8250 | 0.2010 | 2,149,503 | 19m57s |

Table 10: **Ablation of opacity reset strategy.** While resetting low-opacity Gaussians slightly reduces the number of retained Gaussians, it degrades visual quality and destabilizes training.

The primary issue with opacity resetting lies in its tendency to prematurely eliminate informative Gaussians that may exhibit low opacity during intermediate training stages but contribute significantly to final reconstruction quality. Additionally, frequent opacity fluctuations create training instability that disrupts natural convergence patterns, resulting in degraded PSNR (-0.12 dB) and SSIM (-0.0021) performance. This analysis underscores the importance of balanced pruning strategies that prioritize training stability and model robustness over aggressive efficiency gains. The results demonstrate that intelligent, gradual pruning approaches are superior to aggressive reset-based strategies.

## C.3  VGGT ANALYSIS

To further validate that our performance gains stem from architectural contributions rather than prior dependency, we evaluate advanced geometric foundation models (VGGT) (Wang et al., 2025a) as alternative priors. Table 11 compares the performance of GRASP-GS using VGGT priors with our MoGe-based approach.

| Scene | VGGT-based | | | | | MoGe-based | | | | |
|---|---|---|---|---|---|---|---|---|---|---|
| | PSNR↑ | SSIM↑ | LPIPS↓ | GS Num↓ | Time↓ | PSNR↑ | SSIM↑ | LPIPS↓ | GS Num↓ | Time↓ |
| Bicycle | 25.43 | **0.775** | **0.198** | 3.66M | 22m56s | **25.44** | 0.774 | 0.203 | **3.54M** | 22m12s |
| Bonsai | 32.67 | **0.951** | 0.168 | 1.00M | 17m14s | **32.68** | 0.950 | 0.172 | **0.91M** | 16m45s |
| Counter | **29.34** | **0.921** | 0.164 | 0.90M | 19m25s | 29.32 | 0.920 | 0.168 | **0.81M** | 18m45s |
| Flowers | 21.84 | **0.613** | 0.337 | 2.78M | 19m13s | **21.85** | 0.612 | **0.337** | **2.55M** | 19m10s |
| Garden | 27.54 | 0.865 | **0.108** | 3.69M | 23m32s | **27.55** | **0.866** | 0.110 | **3.52M** | 23m16s |
| Kitchen | 31.97 | 0.935 | **0.109** | 0.93M | 20m48s | **32.04** | **0.936** | **0.109** | **0.89M** | 20m14s |
| Room | 32.28 | **0.935** | 0.173 | 0.91M | 19m47s | **32.32** | 0.935 | 0.174 | **0.86M** | 19m44s |
| Stump | 27.03 | 0.790 | **0.209** | 3.83M | 20m28s | **27.05** | **0.791** | 0.210 | **3.50M** | 19m12s |
| Treehill | **22.56** | **0.642** | 0.328 | 2.89M | 19m32s | 22.55 | 0.641 | 0.329 | **2.74M** | 19m26s |
| **Average** | 27.85 | 0.825 | 0.199 | 2.28M | 20m19s | **27.86** | 0.825 | 0.201 | **2.15M** | 19m57s |

Table 11: Performance comparison between VGGT-based and MoGe-based GRASP-GS initialization. Results show similar performance, validating that our method is not dependent on specific prior quality.

The quantitative results demonstrate that VGGT-based initialization achieves nearly identical performance to our MoGe-based approach (27.85 vs 27.86 dB PSNR, 0.825 vs 0.825 SSIM, 0.199 vs

0.201 LPIPS on average). The minimal performance difference further validates that our performance gains stem from architectural contributions (geometric registration, filtering, and dual-stage pruning) rather than dependency on a specific high-quality prior. This observation, combined with our ablation studies showing robustness to weaker priors (Depth Anything), strongly supports that GRASP-GS is an effective architectural design rather than merely leveraging good priors.

## C.4 MULTI-VIEW COMPARISON (MASt3R) ANALYSIS

Comprehensive comparisons are conducted with MASt3R-based multi-view dense priors. To ensure fair comparison, the same pruning strategy and farthest point sampling method are applied to MASt3R-based initialization, selecting 8 views and maintaining approximately 1M initial points. However, MASt3R produces a globally-aligned, fused point cloud that is inherently coupled across views, making it incompatible with our per-view geometric and photometric registration method. Therefore, MASt3R's built-in global alignment approach with COLMAP camera poses is employed. Table 12 presents detailed quantitative results comparing MASt3R-based initialization with our MoGe-based approach across all Mip-NeRF360 scenes.

| Scene | MASt3R-based | | | MASt3R-based | | MoGe-based | | | MoGe-based | |
|---|---|---|---|---|---|---|---|---|---|---|
| | PSNR↑ | SSIM↑ | LPIPS↓ | GS Num↓ | Time↓ | PSNR↑ | SSIM↑ | LPIPS↓ | GS Num↓ | Time↓ |
| Bicycle | 22.80 | 0.731 | 0.219 | **2.13M** | **18m48s** | 25.44 | **0.774** | **0.204** | 3.54M | 22m12s |
| Bonsai | 25.11 | 0.8346 | 0.3027 | **0.78M** | **15m43s** | 32.68 | 0.950 | 0.172 | 0.91M | 16m45s |
| Counter | 28.40 | 0.9122 | 0.1794 | **0.80M** | **18m18s** | 29.32 | 0.920 | 0.168 | 0.81M | 18m45s |
| Flowers | 19.99 | 0.5307 | 0.3786 | **1.75M** | **16m03s** | 21.85 | 0.612 | 0.337 | 2.55M | 16m10s |
| Garden | 26.67 | 0.8458 | 0.123 | 3.78M | 23m25s | 27.55 | 0.866 | 0.110 | **3.52M** | **23m16s** |
| Kitchen | 30.89 | 0.9322 | 0.1138 | 1.13M | 19m48s | 32.04 | 0.936 | 0.109 | **0.89M** | **19m44s** |
| Room | 31.55 | 0.9314 | 0.1788 | 1.21M | 19m57s | 32.32 | 0.936 | 0.174 | **0.86M** | **19m44s** |
| Stump | 23.91 | 0.6906 | 0.2831 | **2.72M** | **18m42s** | 27.05 | 0.791 | 0.210 | 3.50M | 19m12s |
| Treehill | 21.33 | 0.6223 | 0.3067 | 3.26M | 20m10s | 22.55 | 0.641 | 0.329 | **2.74M** | **19m26s** |
| **Average** | 25.63 | 0.780 | 0.234 | **1.95M** | **19m00s** | 27.86 | 0.825 | 0.201 | 2.15M | 19m57s |

Table 12: Performance comparison between MASt3R-based and MoGe-based GRASP-GS initialization across all Mip-NeRF360 scenes.

Comprehensive experimental comparison demonstrates that carefully filtered single-view monocular priors, combined with robust geometric registration using COLMAP poses, achieve superior performance compared to multi-view dense priors (average improvement: +2.23 dB PSNR, +0.045 SSIM, -0.033 LPIPS), with particularly significant improvements in challenging scenes (e.g., Bonsai: +7.57 dB PSNR, Stump: +3.14 dB PSNR, Bicycle: +2.64 dB PSNR). Although multi-view methods like MASt3R are expected to provide better initialization accuracy and multi-view consistency through global alignment, the experimental results show that this theoretical advantage does not lead to superior performance in practice.

To provide direct visual evidence, Figure 12 compares the alignment between COLMAP sparse points (highlighted in yellow) and candidate point clouds from MoGe and MASt3R for the Garden scene. The visualization demonstrates that MASt3R point clouds exhibit significant misalignment with COLMAP points, indicating inconsistency despite theoretical multi-view alignment. In contrast, MoGe-based point clouds, after applying our consistency filtering module, show excellent alignment with COLMAP points. This suggests that the per-view geometric accuracy of MASt3R point clouds is generally lower than that of MoGe, and the multi-view fusion process may introduce errors that compromise the final alignment quality.

To further analyze the quality differences between initialization methods, the initial point clouds for representative scenes (Room and Garden) are compared in Figure 13, including baseline 3D-GS (COLMAP sparse points), MoGe-based initialization, and MASt3R-based initialization. The visualizations reveal that while MASt3R provides denser point clouds, they exhibit significant geometric distortions such as wall warping and contain numerous trailing points (visible as outliers and artifacts), which adversely affect subsequent Gaussian Splatting optimization. In contrast, MoGe-based point clouds, after geometric filtering, exhibit superior per-view geometric accuracy with better alignment to scene structures and fewer outliers. Methods like MASt3R and DUSt3R fundamentally generate point clouds by predicting monocular geometric priors and then performing global alignment or progressive fusion. While this approach provides better theoretical consistency, errors

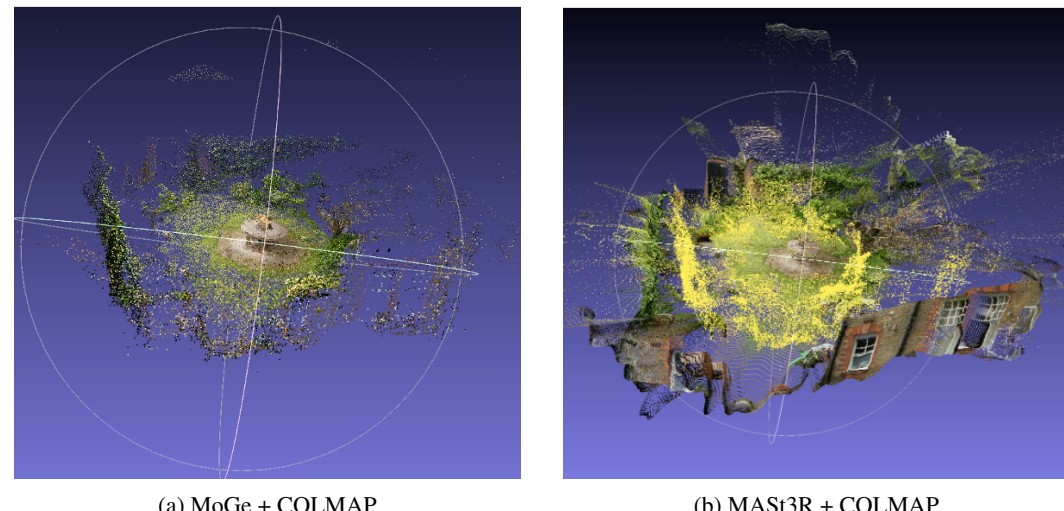

(a) MoGe + COLMAP        (b) MASt3R + COLMAP

Figure 12: Visual comparison of point cloud consistency with COLMAP reference points (highlighted in yellow) for the Garden scene. Left: MoGe-based point clouds after consistency filtering show excellent alignment with COLMAP points. Right: MASt3R point clouds exhibit significant misalignment with COLMAP points, demonstrating inconsistency despite theoretical multi-view alignment.

from individual depth predictions may be propagated and artifacts may be introduced during the fusion process. In contrast, our approach directly leverages high-quality monocular priors and applies sophisticated filtering, offering greater generality and flexibility. The key insight is that point cloud quality and per-view accuracy are more critical than density or multi-view consistency alone, especially when proper geometric alignment and filtering are performed.

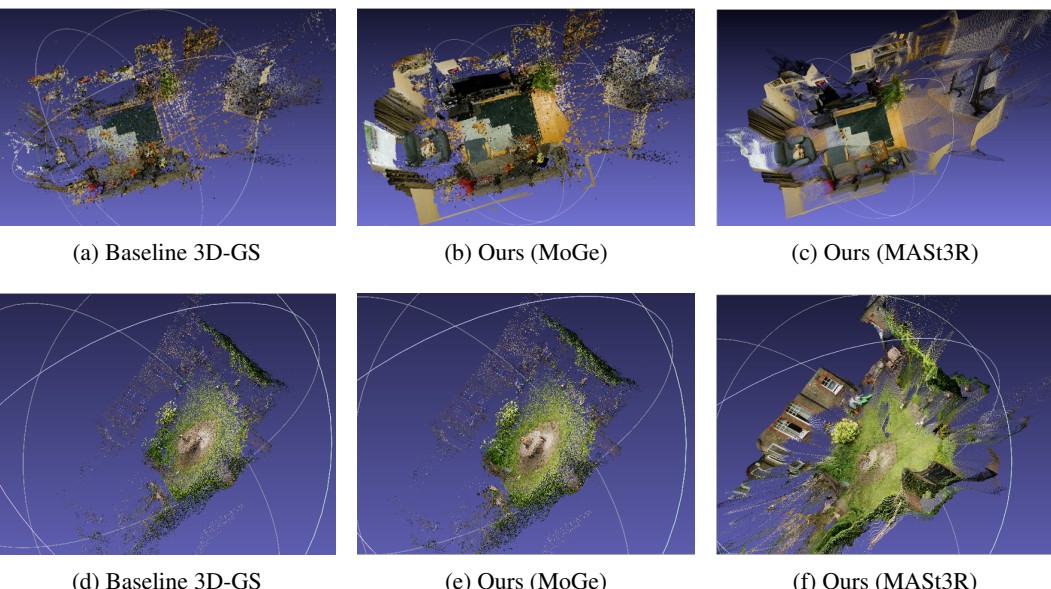

(a) Baseline 3D-GS       (b) Ours (MoGe)       (c) Ours (MASt3R)

(d) Baseline 3D-GS       (e) Ours (MoGe)       (f) Ours (MASt3R)

Figure 13: Comparison of initial point clouds for Room (top row) and Garden (bottom row) scenes. Left: Baseline 3D-GS (COLMAP sparse points); Middle: Our method with MoGe-based initialization; Right: Our method with MASt3R-based initialization.

## D    LLM USAGE STATEMENT

We used large language models (e.g., ChatGPT, Qwen) to assist with grammar checking and language polishing. All content and ideas are authored by the human authors.

