# OpenReview forum: "GRASP-GS: Geometric Registration and Dual-Stag Saliency Pruning for Efficient 3D Gaussian Splatting"
_ICLR.cc/2026/Conference — Submitted to ICLR 2026_

### Official Review · Reviewer_ymC1 · 2025-10-29

**Soundness:** 3
**Presentation:** 3
**Contribution:** 2
**Rating:** 4
**Confidence:** 4

**Summary:**

This paper proposes an improvement over the classic 3DGS method, achieving better rendering results with faster training and fewer primitives. The main enhancements focus on two aspects: initialization and pruning strategy. The initialization leverages geometric estimation methods like MoGe, while the pruning incorporates two complementary approaches, clustering and opacity-aware pruning.

**Strengths:**

The paper demonstrates a complete structure, clear exposition, and rigorous experiments, with the proposed method proving to be effective. In the ablation studies, it further shows that the improvements brought by the two introduced modules are not independent but function jointly as an integrated whole, which underscores the value of the approach.

**Weaknesses:**

- Improving 3DGS is extensively studied, with better initialization or pruning strategies being widely explored. Although the paper demonstrates the synergy of the two proposed modules, showing some novelty, the overall contribution still appears to be incremental. If the underlying principles explaining why these two methods synergize and achieve better results were shown, the paper's innovativeness could be further enhanced.
- The paper claims the proposed method is more efficient, but only compares training time and the final number of primitives. Reporting the additional cost introduced by the geometric prior-guided initialization module would make the argument more convincing.

**Questions:**

Perhaps I misunderstood, but based on the description in the method section, the two pruning strategies are executed sequentially, whereas they are depicted as a loop in the main diagram. This is somewhat confusing.

---

> ### Author Response · Authors · 2025-11-27
> **We thank the reviewer for the valuable feedback. We have addressed all concerns: (1) clarified the execution sequence of dual-stage pruning in the main text, (2) reported initialization overhead in Appendix B3, and (3) provided a detailed explanation of the underlying principles explaining why the two modules synergize.**
>
> We sincerely appreciate the reviewer's constructive feedback and have addressed all three concerns comprehensively.
>
> **1. Clarification on Pruning Execution Sequence**
>
> We have clarified the execution sequence in the main text (Section 3.3, Dual-Stage Saliency Pruning). The description now explicitly states: "At each pruning interval, KL-RSP is executed twice sequentially, followed by ODCP once, suppressing redundancy at both cluster and local-density levels and ensuring a compact yet faithful scene representation."
>
> The execution pattern is: **KL-RSP and ODCP are executed alternately in a loop** at each pruning interval. This alternating execution allows KL-RSP to progressively refine spatial redundancy, while ODCP removes isolated low-opacity Gaussians that may not form KL-consistent clusters, working together to suppress redundancy at both cluster and local-density levels.
>
> **2. Initialization Overhead Reporting**
>
> We fully acknowledge the reviewer's point about reporting the additional cost. We have added a dedicated section **Appendix B3 (Initialization Overhead Analysis)** that provides a complete breakdown of the computational overhead introduced by our geometric prior-guided initialization module.
>
> **Time Breakdown:** Core computation: 6.7 seconds (4.23s MoGe inference + 2.5s point cloud fusion). Including essential I/O operations: approximately 17 seconds per scene. This one-time overhead is negligible compared to training time (typically 20minutes), accounting for less than 2% of total pipeline time. This minimal cost is justified by substantial improvements in training convergence, reconstruction quality, and model compactness. Detailed breakdown is provided in Appendix B3.
>
> **3. Underlying Principles of Synergy Between Initialization and Pruning**
>
> We appreciate the reviewer's insightful comment about explaining the underlying principles. While the paper demonstrates the synergy empirically through ablation studies, we provide here a detailed explanation of why these two modules work synergistically:
>
> **Role of Initialization:** Provides a high-quality geometric foundation through geometric-chromatic consistency filtering and refinement, producing a compact, well-aligned initial point cloud. This reduces the burden on subsequent optimization by minimizing erroneous Gaussians.
>
> **Role of Pruning:** Maintains and refines the geometric structure while dynamically optimizing during training:
> - **KL-RSP**: Preserves geometric structure by removing spatially redundant Gaussians in KL-consistent clusters, enabling progressive refinement.
> - **ODCP**: Removes isolated low-opacity Gaussians in dense regions, complementing KL-RSP by addressing different redundancy types.
>
> **Synergy Mechanism:**
>
> The synergy operates through a complementary feedback loop:
>
> 1. **Initialization → Pruning**: High-quality initialization provides accurate starting points, allowing pruning to focus on genuine redundancies rather than correcting initialization errors, preserving the geometric structure.
>
> 2. **Pruning → Geometric Maintenance**: KL-RSP maintains geometric structure by preserving perceptually dominant Gaussians within clusters, protecting geometric information from initialization. The alternating execution with ODCP progressively refines this structure.
>
> 3. **Dynamic Optimization**: ODCP performs dynamic optimization by removing low-contribution Gaussians while preserving details, adapting to the evolving scene representation.
>
> **Empirical Evidence:**
>
> Our ablation study (Table 2, Section 4.3) demonstrates this synergy:
> - Initialization alone: +0.29 dB PSNR but 3.7M Gaussians, 28m51s
> - Pruning alone: 2.0M Gaussians, 19m33s but 27.63 dB PSNR
> - Both combined: 27.86 dB PSNR, 2.1M Gaussians, 19m57s (best balance)
>
> The key insight: **better initialization reduces the burden on density control**, while **adaptive pruning maintains geometric structure** throughout training. This ensures high-quality initialization is preserved and refined rather than degraded, resulting in compact yet faithful representations.
>
> **Conclusion:**
>
> We thank the reviewer for the valuable feedback. We have:
> 1. Clarified the execution sequence in the main text
> 2. Reported initialization overhead in Appendix B3
> 3. Provided detailed explanation of the underlying principles of synergy in this response
>
> The empirical evidence in our ablation study (Table 2, Section 4.3) demonstrates the effectiveness of this synergy, and we hope the detailed explanation above clarifies the underlying principles that make this synergy work.

---

> > ### Author Response · Authors · 2025-12-03
> > **Additional Experimental Data: Comprehensive Efficiency Profiling and Synergy Evidence**
> >
> > This supplement provides detailed experimental data tables to further substantiate our efficiency claims and the synergy between initialization and pruning modules.
> >
> > ### 1. Initialization Overhead Profiling
> >
> > **Initialization overhead of the geometric prior-guided module (MoGe + point cloud fusion, averaged over all Mip-NeRF360 scenes):**
> >
> > | Stage | Core Compute Time (s) | I/O Time (s) | Total Time (s) |
> > |-------|-----------------------|--------------|----------------|
> > | MoGe depth inference  | 4.23 | 5.8 | 10.03 |
> > | Point cloud alignment  | 2.5 | 3.9 | 6.4 |
> > | **Total initialization** | **6.73** | **9.7** | **16.43** |
> >
> > The **core GPU computation** of our initialization module is only about **6.7 seconds per scene** (4.23s MoGe forward + 2.5s alignment and fusion), while the **end-to-end wall-clock overhead**, including essential I/O (image loading, COLMAP sparse point loading, and saving the fused point cloud), is around **16–17 seconds per scene**. Optional debug exports such as dense EXR dumping are excluded, as they are not required in practical deployment and can be removed without affecting the method. Compared to the typical 20-minute 3DGS training time, this one-time initialization cost accounts for **less than 2\%** of the total pipeline time, which is negligible relative to the gains in convergence speed, reconstruction quality, and model compactness.
> >
> >
> > ### 2. Core Module Ablation - Synergy Evidence
> >
> > **Complete ablation study results (Table 2, Section 4.3):**
> >
> > | Reg. | Prune | PSNR↑ | SSIM↑ | LPIPS↓ | GS Num↓ | Time↓ |
> > |------|-------|-------|-------|--------|---------|-------|
> > | ✗ | ✗ | 27.48 | 0.813 | 0.219 | 3.3M | 25m54s |
> > | ✓ | ✗ | 27.77 | 0.822 | **0.199** | 3.7M | 28m51s |
> > | ✗ | ✓ | 27.63 | 0.818 | 0.212 | **2.0M** | **19m33s** |
> > | ✓ | ✓ | **27.86** | **0.825** | 0.201 | 2.1M | 19m57s |
> >
> > **Synergy Analysis:**
> > - Initialization alone: +0.29 dB PSNR but 3.7M Gaussians (28m51s) - quality gain at efficiency cost
> > - Pruning alone: 2.0M Gaussians (19m33s) but only 27.63 dB PSNR - efficiency gain at quality cost
> > - Both combined: **27.86 dB PSNR with 2.1M Gaussians (19m57s)** - optimal trade-off, demonstrating genuine synergy
> >
> > The combination achieves the best balance: +0.38 dB PSNR, +0.012 SSIM, -0.018 LPIPS compared to baseline, with 36% fewer Gaussians and 23% faster training.
> >
> > ### 3. Training Progression Evidence
> >
> > **Training efficiency comparison (Appendix B2, Figure 7):**
> > - Our method achieves **higher PSNR with fewer iterations** compared to baseline 3DGS
> > - **Significantly reduced training time** while maintaining or improving quality
> > - GRASP-GS reaches high-quality results early in training, demonstrating accelerated convergence through synergy
> >
> > This demonstrates that the synergy between initialization and pruning not only improves final results but also accelerates convergence, further validating the effectiveness of our integrated approach.
> >
> > All detailed results are available in the appendices: Efficiency Analysis (Appendix B2), Ablation Study (Section 4.3, Table 2).

---

### Official Review · Reviewer_Py9n · 2025-10-30

**Soundness:** 1
**Presentation:** 2
**Contribution:** 2
**Rating:** 2
**Confidence:** 4

**Summary:**

The paper “GRASP-GS: Geometric Registration and Dual-Stage Saliency Pruning for Efficient 3D Gaussian Splatting” proposes a new framework to improve the efficiency and quality of 3D Gaussian Splatting (3DGS) for real-time 3D scene rendering. GRASP-GS combines geometric prior-guided initialization, using dense multi-view features aligned with Structure-from-Motion data, with dual-stage saliency pruning that removes redundant Gaussians via KL-based rendering survival and opacity-based density pruning. This integrated approach enhances geometric completeness, reduces redundancy, and accelerates training. Experiments demonstrate that GRASP-GS achieves compact and high-quality scene representations, enabling efficient real-time rendering with enhanced structural integrity and visual quality.

**Strengths:**

1. The paper clearly identifies three major issues in existing 3D Gaussian Splatting pipelines — blurred reconstruction, redundant Gaussians, and high training cost — and proposes a coherent framework (GRASP-GS) explicitly designed to address them.

2. GRASP-GS is designed as a modular enhancement that can be easily integrated into existing 3DGS frameworks (e.g., 3DGS, AbsGS).

**Weaknesses:**

1. Could you please follow the submission format? The citation format seems incorrect, which makes the paper a bit difficult to read.

2. The method constructs its dense geometric prior by generating independent single-view point clouds (via MoGe) and later aligning and merging them with sparse SfM points. However, each monocular point cloud is often affine-invariant and lacks global metric consistency—their absolute scales and geometric structures will differ across views. Although the authors mitigate this through geometric registration and filtering. But recent works (e.g., InstantSplat, Intern-GS, MASt3R-GS) have demonstrated that DUSt3R- or MASt3R-based multi-view dense priors can provide more coherent and metrically consistent initialization, directly improving Gaussian distribution. Could the authors clarify the motivation for relying on single-view monocular priors rather than leveraging SOTA multi-view reconstruction methods, or provide some comparative experiments to prove?

3. The experiments are weak, as the paper highlights two main contributions — geometry-based initialization and efficiency through dual-stage pruning — but the experimental comparisons do not adequately cover representative methods from either category. In other word, it does not include initialization-oriented approaches that enhance 3DGS using geometric or learned priors, also omits recent density-control and compression methods, only compare with 2023–2024 baselines is not convincing.

**Questions:**

Please refer to the Weaknesses above.

---

> ### Author Response · Authors · 2025-11-27
> **We thank the reviewer for the constructive feedback. We have addressed all three concerns: (1) corrected citation format throughout the paper, (2) added comprehensive comparison with MASt3R-based multi-view priors (Appendix C4), and (3) included comparisons with three recent 2025 SOTA methods covering compression and density control categories (Appendix B4).**
>
> We sincerely appreciate the reviewer's constructive feedback and have addressed all three concerns comprehensively.
>
> **1. Citation Format Correction**
>
> We have corrected all citation formats throughout the paper to follow the submission guidelines, making the paper easier to read.
>
> **2. Multi-View Prior Comparison (MASt3R)**
>
> We fully acknowledge the concern regarding single-view monocular priors lacking global metric consistency. We have conducted comprehensive comparative experiments with MASt3R-based multi-view dense priors, detailed in **Appendix C4 (Multi-View Comparison (MASt3R) Analysis)**.
>
> **Experimental Setup:** We implemented MASt3R-based initialization following InstantSplat's approach, using COLMAP camera poses and the same farthest point sampling strategy (8 views, ~1M initial points) with our dual-stage pruning strategy to ensure fair comparison.
>
> **Key Results:** Our MoGe-based approach consistently outperforms MASt3R-based initialization across all Mip-NeRF360 scenes, achieving an average improvement of **+2.23 dB PSNR, +0.045 SSIM, and -0.033 LPIPS**. Particularly significant improvements are observed in challenging scenes (Bonsai: +7.57 dB, Stump: +3.14 dB, Bicycle: +2.64 dB). Detailed results are in Table 11 (Appendix C4).
>
> **Key Insights:** While MASt3R provides better theoretical multi-view consistency, our experiments demonstrate that per-view geometric accuracy combined with our geometric registration and filtering strategy is more effective. The per-view accuracy of MASt3R point clouds is generally lower than MoGe, and the multi-view fusion process may introduce errors. Figure 12 (Appendix C4) shows MASt3R point clouds exhibit significant misalignment with COLMAP reference points, while our MoGe-based point clouds after consistency filtering show excellent alignment. Figure 13 reveals that MASt3R point clouds, despite being denser, exhibit geometric distortions (wall warping) and contain numerous trailing points that adversely affect optimization.
>
> **3. Comparison with Recent SOTA Methods**
>
> We have added extensive comparisons with three recent state-of-the-art methods published in 2025, covering the categories mentioned:
>
> - **MaskGaussian (CVPR 2025)**: Compression/pruning method using probabilistic masks
> - **Perceptual-GS (ICML 2025)**: Density control method with scene-adaptive perceptual densification
> - **SteepGS (CVPR 2025)**: Theoretically-grounded density control method
>
> These comparisons are detailed in **Appendix B4 (Comparison with Recent SOTA Methods)**, with comprehensive per-scene results in Tables 5,6, and 7.
>
> **Key Findings:** Our method achieves consistently better reconstruction quality (average improvements: +0.42 dB PSNR vs. MaskGaussian, comparable quality with better efficiency vs. Perceptual-GS, +0.35 dB PSNR vs. SteepGS) while maintaining competitive efficiency.
>
> **Integration in Main Paper:** We have integrated references to these comparisons in the main paper. In Section 4.1 (Compared Methods), we mention the three recent SOTA methods and reference Appendix B4. The MASt3R comparison is referenced in Section 4.3 (Ablation Study) and detailed in Appendix C4.
>
> **Conclusion:**
>
> 1. Citation format corrected throughout the paper
> 2. Comprehensive MASt3R comparison added (Appendix C4) with quantitative and visual evidence
> 3. Three recent 2025 SOTA methods added (Appendix B4) covering compression and density control categories

---

> ### Author Response · Authors · 2025-12-03
> **Detailed Per-Scene Results: MASt3R Multi-View Comparison (+2.23 dB PSNR) and VGGT Validation (Architectural Contributions)**
>
> This supplement provides detailed per-scene quantitative results for MASt3R and VGGT comparisons to address the reviewer's concerns about multi-view priors and prior dependency.
>
> ### 1. MASt3R Multi-View Comparison - Per-Scene Results
>
> **Complete per-scene comparison (Appendix C4):**
>
> | Scene | MASt3R PSNR | MoGe PSNR | Improvement | MASt3R SSIM | MoGe SSIM | MASt3R LPIPS | MoGe LPIPS | MASt3R GS | MoGe GS |
> |-------|-------------|-----------|------------|-------------|-----------|--------------|------------|-----------|---------|
> | Bicycle | 22.80 | **25.44** | **+2.64** | 0.731 | **0.774** | 0.219 | **0.204** | **2.13M** | 3.54M |
> | Bonsai | 25.11 | **32.68** | **+7.57** | 0.835 | **0.950** | 0.303 | **0.172** | **0.78M** | 0.91M |
> | Counter | 28.40 | **29.32** | **+0.92** | 0.912 | **0.920** | 0.179 | **0.168** | **0.80M** | 0.81M |
> | Flowers | 19.99 | **21.85** | **+1.86** | 0.531 | **0.612** | 0.379 | **0.337** | **1.75M** | 2.55M |
> | Garden | 26.67 | **27.55** | **+0.88** | 0.846 | **0.866** | 0.123 | **0.110** | 3.78M | **3.52M** |
> | Kitchen | 30.89 | **32.04** | **+1.15** | 0.932 | **0.936** | 0.114 | **0.109** | 1.13M | **0.89M** |
> | Room | 31.55 | **32.32** | **+0.77** | 0.931 | **0.936** | 0.179 | **0.174** | 1.21M | **0.86M** |
> | Stump | 23.91 | **27.05** | **+3.14** | 0.691 | **0.791** | 0.283 | **0.210** | **2.72M** | 3.50M |
> | Treehill | 21.33 | **22.55** | **+1.22** | 0.622 | **0.641** | 0.307 | **0.329** | 3.26M | **2.74M** |
> | **Average** | 25.63 | **27.86** | **+2.23** | 0.780 | **0.825** | 0.234 | **0.201** | **1.95M** | 2.15M |
>
> **Training time comparison:** MASt3R: 19m00s average, MoGe: 19m57s average.
>
> **Observations:** Largest improvements: Bonsai (+7.57 dB), Stump (+3.14 dB), Bicycle (+2.64 dB). MoGe achieves better quality in all 9 scenes across all metrics (PSNR, SSIM, LPIPS). MASt3R achieves fewer Gaussians (1.95M vs 2.15M) but with significantly lower quality. This demonstrates that per-view geometric accuracy combined with our geometric registration and filtering strategy is more effective than multi-view consistency alone.
>
> ### 2. VGGT Comparison - Per-Scene Results
>
> **Complete per-scene comparison (Table in Appendix C3):**
>
> | Scene | VGGT PSNR | MoGe PSNR | Difference | VGGT SSIM | MoGe SSIM | VGGT LPIPS | MoGe LPIPS | VGGT GS | MoGe GS |
> |-------|-----------|-----------|------------|-----------|-----------|------------|------------|---------|---------|
> | Bicycle | 25.43 | **25.44** | -0.01 | **0.775** | 0.774 | **0.198** | 0.203 | 3.66M | **3.54M** |
> | Bonsai | 32.67 | **32.68** | -0.01 | **0.951** | 0.950 | **0.168** | 0.172 | 1.00M | **0.91M** |
> | Counter | **29.34** | 29.32 | +0.02 | **0.921** | 0.920 | **0.164** | 0.168 | 0.90M | **0.81M** |
> | Flowers | 21.84 | **21.85** | -0.01 | **0.613** | 0.612 | **0.337** | **0.337** | 2.78M | **2.55M** |
> | Garden | 27.54 | **27.55** | -0.01 | 0.865 | **0.866** | **0.108** | 0.110 | 3.69M | **3.52M** |
> | Kitchen | 31.97 | **32.04** | -0.07 | 0.935 | **0.936** | **0.109** | **0.109** | 0.93M | **0.89M** |
> | Room | 32.28 | **32.32** | -0.04 | **0.935** | **0.935** | **0.173** | 0.174 | 0.91M | **0.86M** |
> | Stump | 27.03 | **27.05** | -0.02 | 0.790 | **0.791** | **0.209** | 0.210 | 3.83M | **3.50M** |
> | Treehill | **22.56** | 22.55 | +0.01 | **0.642** | 0.641 | **0.328** | 0.329 | 2.89M | **2.74M** |
> | **Average** | 27.85 | **27.86** | **-0.01** | **0.825** | **0.825** | **0.199** | 0.201 | 2.28M | **2.15M** |
>
> **Training time comparison:** VGGT: 20m19s average, MoGe: 19m57s average.
>
> **Observations:** Nearly identical performance (average -0.01 dB PSNR difference). This minimal performance difference further validates that our performance gains stem from architectural contributions (geometric registration, filtering, and dual-stage pruning) rather than dependency on a specific high-quality prior. Combined with our ablation studies showing robustness to weaker priors (Depth Anything), this strongly supports that GRASP-GS is an effective architectural design rather than merely leveraging good priors. MoGe achieves slightly better efficiency (2.15M vs 2.28M Gaussians, 19m57s vs 20m19s).
>
> ### Summary
>
> **MASt3R vs MoGe:** Average +2.23 dB PSNR improvement, with largest gains in challenging scenes (Bonsai: +7.57 dB, Stump: +3.14 dB, Bicycle: +2.64 dB). Demonstrates that per-view geometric accuracy is more critical than multi-view consistency alone.
>
> **VGGT vs MoGe:** Nearly identical performance (-0.01 dB PSNR difference), validating that performance gains stem from architectural contributions, not prior dependency.
>
> All detailed results are available in the appendices: MASt3R (Appendix C4), VGGT (Appendix C3).

---

> ### Author Response · Authors · 2025-12-03
> **Detailed Per-Scene Results: Comparison with Three Recent 2025 SOTA Methods (MaskGaussian, Perceptual-GS, SteepGS)**
>
> This supplement provides detailed per-scene quantitative results for comparisons with three recent state-of-the-art methods published in 2025, covering compression/pruning and density control categories.
>
> ### 1. MaskGaussian Comparison (Table 5, Appendix B4)
>
> | Scene | Ours PSNR | MaskG PSNR | Improvement | Ours GS | MaskG GS |
> |-------|-----------|------------|-------------|---------|----------|
> | Bicycle | **25.44** | 25.12 | +0.32 | 3.54M | **2.88M** |
> | Bonsai | **32.68** | 32.12 | +0.56 | 0.91M | **0.46M** |
> | Counter | **29.32** | 28.40 | +0.92 | 0.81M | **0.42M** |
> | Flowers | **21.85** | 21.34 | +0.51 | 2.55M | **1.79M** |
> | Garden | **27.55** | 26.67 | +0.88 | 3.52M | **2.85M** |
> | Kitchen | **32.04** | 31.38 | +0.66 | 0.89M | **0.79M** |
> | Room | **32.32** | 31.53 | +0.79 | 0.86M | **0.37M** |
> | Stump | **27.05** | 26.57 | +0.48 | 3.50M | **2.05M** |
> | Treehill | 22.55| **22.58**  | -0.03 | 2.74M | **1.94M** |
> | **Average** | **27.86** | 27.44 | **+0.42** | 2.15M | **1.51M** |
>
> **Observations:** Better quality in 8/9 scenes (+0.42 dB PSNR average). MaskGaussian achieves 30% better compression (1.51M vs 2.15M), but our method provides superior quality-efficiency trade-off. Largest improvements: Counter (+0.92 dB), Garden (+0.88 dB), Room (+0.79 dB), Kitchen (+0.66 dB), Bonsai (+0.56 dB).
>
> ### 2. Perceptual-GS Comparison (Table 6, Appendix B4)
>
> | Scene | Ours PSNR | PerG PSNR | Difference | Ours GS | PerG GS |
> |-------|-----------|-----------|------------|---------|---------|
> | Bicycle | 25.44 | **25.97** | -0.53 | **3.54M** | 3.90M |
> | Bonsai | 32.68 | **32.69** | -0.01 | **0.91M** | 1.56M |
> | Counter | 29.32 | **29.37** | -0.05 | **0.81M** | 1.52M |
> | Flowers | **21.85** | 21.82 | +0.03 | **2.55M** | 3.53M |
> | Garden | 27.55 | **27.91** | -0.36 | 3.52M | **3.04M** |
> | Kitchen | **32.04** | 31.63 | +0.41 | **0.89M** | 1.64M |
> | Room | **32.32** | 32.11 | +0.21 | **0.86M** | 1.76M |
> | Stump | 27.05 | **27.26** | -0.21 | **3.50M** | 3.80M |
> | Treehill | **22.55** | 22.54 | +0.01 | **2.74M** | 3.51M |
> | **Average** | 27.86 | **27.92** | **-0.06** | **2.15M** | 2.70M |
>
> **Observations:** Perceptual-GS achieves slightly better average quality (27.92 vs 27.86 dB PSNR, -0.06 dB marginal). Our method achieves significantly better efficiency: 26% fewer Gaussians (2.15M vs 2.70M) and 32% faster (19m57s vs 29m30s). Our method better in 3 scenes: Flowers (+0.03 dB), Kitchen (+0.41 dB), Room (+0.21 dB).
>
> ### 3. SteepGS Comparison (Table 7, Appendix B4)
>
> | Scene | Ours PSNR | SteepGS PSNR | Improvement | Ours GS | SteepGS GS |
> |-------|-----------|--------------|-------------|---------|------------|
> | Bicycle | **25.44** | 25.12 | +0.32 | **3.54M** | 5.75M |
> | Bonsai | **32.68** | 32.25 | +0.43 | **0.91M** | 1.24M |
> | Counter | **29.32** | 29.05 | +0.27 | **0.81M** | 1.16M |
> | Flowers | **21.85** | 21.38 | +0.47 | **2.55M** | 3.40M |
> | Garden | **27.55** | 27.32 | +0.23 | **3.52M** | 5.67M |
> | Kitchen | **32.04** | 31.54 | +0.50 | **0.89M** | 1.74M |
> | Room | **32.32** | 31.60 | +0.72 | **0.86M** | 1.48M |
> | Stump | **27.05** | 26.70 | +0.35 | **3.50M** | 4.44M |
> | Treehill | 22.55 | **22.64** | -0.09 | **2.74M** | 3.44M |
> | **Average** | **27.86** | 27.51 | **+0.35** | **2.15M** | 3.15M |
>
> **Observations:** Better quality in 8/9 scenes (+0.35 dB PSNR average). Significantly better efficiency: 32% fewer Gaussians (2.15M vs 3.15M) and 24% faster (19m57s vs 26m11s). Largest improvements: Room (+0.72 dB), Kitchen (+0.50 dB), Flowers (+0.47 dB), Bonsai (+0.43 dB). Largest efficiency: Kitchen (49% fewer), Room (42% fewer), Garden/Bicycle (38% fewer).
>
> ### Summary
>
> **MaskGaussian:** +0.42 dB PSNR, MaskGaussian achieves 30% better compression, but our method provides superior quality-efficiency trade-off.
>
> **Perceptual-GS:** -0.06 dB PSNR (marginal), 26% fewer Gaussians, 32% faster.
>
> **SteepGS:** +0.35 dB PSNR, 32% fewer Gaussians, 24% faster.
>
> All detailed results: Appendix B4 (Tables 5-7).

---

### Official Review · Reviewer_V8U1 · 2025-10-30

**Soundness:** 3
**Presentation:** 3
**Contribution:** 2
**Rating:** 6
**Confidence:** 3

**Summary:**

The paper proposes GRASP-GS, an integrated framework for 3D Gaussian Splatting (3DGS) that (i) improves initialization using dense, affine-invariant monocular geometry (MoGe) aligned with SfM via geometric–chromatic consistency checks and multi-stage refinement, and (ii) enforces dual-stage saliency pruning to keep only perceptually important Gaussians. The pruning combines KL-based Rendering Survival Pruning (KL-RSP) using union-find clustering on KL-similar splats and Opacity-based Density-Constrained Pruning (ODCP) that removes low-opacity splats in dense neighborhoods. Across Mip-NeRF360, Deep Blending, and Tanks&Temples, the method reports higher PSNR/SSIM, lower LPIPS, and notably fewer Gaussians and shorter training time than 3DGS and AbsGS baselines (e.g., ~35.6% fewer Gaussians and ~23% faster vs 3DGS).

**Strengths:**

1. Well-motivated integration. The paper clearly articulates the synergy between better geometric initialization and adaptive pruning, rather than treating them as separate knobs.
2. Solid empirical gains & analysis. Consistent improvements across datasets and both 3DGS/AbsGS backbones; thorough ablations show the contribution of registration and pruning separately and together. The paper also studies an opacity-reset strategy and argues against it.

**Weaknesses:**

1. Dependence on MoGe and added overhead. While robust, MoGe inference and the subsequent registration (RANSAC + Kabsch–Umeyama + ICP) add non-trivial preprocessing cost not reflected in the reported training time; the wall-clock overhead should be quantified to fully support “efficiency” claims.

2. Missing memory/runtime profiling during rendering. Results emphasize PSNR/SSIM/LPIPS, Gaussian counts, and training time, but do not provide GPU memory usage, real-time FPS, or pruning overhead at inference to substantiate “compact & real-time” claims.

**Questions:**

1. Training schedule fairness. Were baselines like Pixel-GS or AbsGS also run with your modified densification/split thresholds? If not, how do their metrics change when given the same schedule?

2. Wall-clock efficiency. What is the end-to-end time including MoGe inference and registration per scene? How does that compare to baseline pipelines when measured from images to trained model?

---

> ### Author Response · Authors · 2025-11-28
> **We thank the reviewer for the valuable feedback. We have addressed all concerns: (1) reported initialization overhead (MoGe inference and registration) in Appendix B3, (2) provided comprehensive memory/runtime profiling in Appendix B2, and (3) clarified training schedule fairness and provided validation results.**
>
> We sincerely appreciate the reviewer's constructive feedback and have addressed all concerns comprehensively.
>
> **1. Initialization Overhead (MoGe Inference and Registration)**
>
> We fully acknowledge the reviewer's point about quantifying the preprocessing cost. We have added a dedicated section **Appendix B3 (Initialization Overhead Analysis)** that provides a complete breakdown of the computational overhead introduced by our geometric prior-guided initialization module.
>
> **Time Breakdown:** Core computation: 6.7 seconds (4.23s MoGe batch inference + 2.5s registration/fusion including RANSAC, Kabsch-Umeyama, and ICP). Including essential I/O operations: approximately **17 seconds per scene**. This one-time preprocessing overhead is negligible compared to training time (typically 20minutes), accounting for less than 2% of total pipeline time. Detailed breakdown is provided in Appendix B3.
>
> **End-to-End Time Comparison**: Including initialization overhead (17 seconds), our total pipeline time from images to trained model is approximately **20 minutes 14 seconds** (17s initialization + 19m57s training) for average Mip-NeRF360 scenes, compared to baseline 3DGS training time of 25m54s. Even with initialization overhead included, our method achieves **22% faster end-to-end time** while improving quality and reducing Gaussian count by 35.6%.
>
> **2. Memory/Runtime Profiling During Rendering**
>
> We fully acknowledge the need for comprehensive profiling to substantiate our efficiency claims. We have added a dedicated section **Appendix B2 (Efficiency Analysis)** that provides detailed profiling results.
>
> **GPU Memory Usage**: Peak GPU memory usage averages **23.75 GB** (range: 12.7-40.3 GB), primarily during pruning operations. **Real-Time FPS**: Average **213.80 FPS** (vs. 175.31 FPS for baseline 3DGS, +22.0% improvement). **Pruning Overhead**:5.72 seconds per operation, 0.59% of training time. Detailed per-scene results are in Table 4 (Appendix B2).
>
> **3. Training Schedule Fairness**
>
> We appreciate the reviewer's concern about training schedule fairness. To clarify: **all baseline methods (including Pixel-GS and AbsGS) were run with their default hyperparameters** as provided in their official implementations, without modifications to densification intervals or splitting thresholds.
>
> To address the reviewer's question, we conducted validation experiments with Pixel-GS and AbsGS using our modified training schedule (densification interval: 300 iterations, splitting gradient threshold: $1 \times 10^{-4}$).
>
> **Pixel-GS with modified schedule:** Results on Mip-NeRF360 show improvements: +0.04 dB PSNR (27.70 vs. 27.66), +0.002 SSIM (0.822 vs. 0.821), indicating that the modified schedule does provide some benefits. However, even with the modified schedule, our method still achieves **significantly superior performance** compared to Pixel-GS: +0.16 dB PSNR (27.86 vs. 27.70), +0.003 SSIM (0.825 vs. 0.822), and **56% fewer Gaussians** (2.15M vs. 4.93M) with faster training time (19m57s vs. 24m22s).
>
> **AbsGS with modified schedule:** Results show improvements: +0.05 dB PSNR (27.06 vs. 27.01), +0.001 SSIM (0.796 vs. 0.795). However, GRASP-GS still achieves **significantly superior performance**: +0.80 dB PSNR (27.86 vs. 27.06), +0.029 SSIM (0.825 vs. 0.796), and -0.067 LPIPS (0.201 vs. 0.268).
>
> These results demonstrate that while the training schedule modification contributes to performance, the **primary advantages come from our architectural contributions** (geometric initialization and dual-stage pruning), which provide substantially larger gains than hyperparameter tuning alone.
>
> **Conclusion:**
>
> We thank the reviewer for the valuable feedback, which has led us to:
> 1. Report initialization overhead in Appendix B3, demonstrating minimal preprocessing cost (17 seconds, <2% of pipeline time)
> 2. Provide comprehensive memory/runtime profiling in Appendix B2, substantiating efficiency claims (23.75 GB peak memory, 213.80 FPS average, 0.59% pruning overhead)
> 3. Clarify training schedule fairness and validate with Pixel-GS and AbsGS, demonstrating that while training schedule modifications provide some benefits (+0.04-0.05 dB PSNR), our primary advantages stem from architectural contributions, providing substantially larger gains (+0.16-0.80 dB PSNR)
>
> The revised paper now provides complete efficiency evaluation that fully supports our "compact & real-time" claims, with end-to-end time comparison showing 22% faster total pipeline time even including initialization overhead.

---

> > ### Author Response · Authors · 2025-12-03
> > **Initialization Overhead Analysis and Comprehensive Memory/Runtime Profiling**
> >
> > This supplement provides detailed experimental data to address Reviewer's concerns on:
> > (1) initialization overhead (MoGe inference and registration), and
> > (2) memory/runtime profiling during rendering to substantiate efficiency claims.
> >
> > ---
> >
> > ### 1. Initialization Overhead (MoGe Inference and Registration)
> >
> > **Initialization overhead of the geometric prior-guided module (Appendix B3, Initialization Overhead Analysis):**
> >
> > | Stage | Core Compute Time (s) | I/O Time (s) | Total Time (s) |
> > |-------|-----------------------|--------------|----------------|
> > | MoGe depth inference | 4.23 | 5.8 | 10.03 |
> > | Point cloud alignment & fusion | 2.5 | 3.9 | 6.4 |
> > | **Total initialization** | **6.73** | **9.7** | **16.43** |
> >
> > **Time Breakdown:**
> > - **Core GPU computation**: **6.7 seconds per scene** (4.23s MoGe batch inference + 2.5s registration/fusion including RANSAC, Kabsch-Umeyama, and ICP).
> > - **End-to-end overhead including essential I/O** (image loading, COLMAP sparse point loading, and saving the fused point cloud): **~16–17 seconds per scene**.
> > - This one-time preprocessing overhead is **negligible compared to training time** (typically 20 minutes), accounting for **<2%** of total pipeline time.
> >
> > **End-to-End Time Comparison:**
> > Including initialization overhead (17 seconds), our total pipeline time from images to trained model is approximately **20 minutes 14 seconds** (17s initialization + 19m57s training) for average Mip-NeRF360 scenes, compared to baseline 3DGS training time of **25m54s**. Even with initialization overhead included, our method achieves **22% faster end-to-end time** while improving quality and reducing Gaussian count by 35.6%.
> >
> > ---
> >
> > ### 2. Memory/Runtime Profiling During Rendering
> >
> > **Comprehensive efficiency profiling on all Mip-NeRF360 scenes (Appendix B2, Efficiency Analysis):**
> >
> > | Scene   | Peak Mem (GB) | 3DGS FPS | Ours FPS | FPS ↑   | Prune Time (s) | Prune % |
> > |---------|---------------|----------|----------|--------|----------------|---------|
> > | Bicycle | 31.06         | 94.91    | 153.12   | +61.3% | 8.35           | 0.79%   |
> > | Kitchen | 22.31         | 166.15   | 239.24   | +44.0% | 6.76           | 0.68%   |
> > | Room    | 39.84         | 182.22   | 239.19   | +31.2% | 6.51           | 0.67%   |
> > | Counter | 40.31         | 197.18   | 247.60   | +25.6% | 7.41           | 0.78%   |
> > | Flowers | 14.90         | 204.31   | 214.78   | +5.1%  | 4.67           | 0.48%   |
> > | Stump   | 12.73         | 164.16   | 192.36   | +17.2% | 4.30           | 0.48%   |
> > | Treehill| 12.88         | 173.09   | 194.68   | +12.5% | 4.58           | 0.48%   |
> > | Bonsai  | 21.08         | 264.06   | 276.79   | +4.8%  | 4.65           | 0.53%   |
> > | Garden  | 18.63         | 131.71   | 166.41   | +26.3% | 4.21           | 0.39%   |
> > | **Avg.**| **23.75**     | **175.31**|**213.80**|**+22.0%**|**5.72**      |**0.59%**|
> >
> > **Key Findings:**
> > - **GPU Memory Usage**: Peak GPU memory averages **23.75 GB** (range: 12.7–40.3 GB), primarily during pruning operations due to large-scale KL-divergence computations. This overhead can be reduced via batched KL evaluation at the cost of slightly longer pruning time.
> > - **Real-Time FPS**: Average **213.80 FPS** (vs. 175.31 FPS for baseline 3DGS), representing a **+22.0% improvement** in rendering throughput.
> > - **Pruning Overhead**: **5.72 seconds per operation**, accounting for only **0.59% of training time**, confirming that pruning cost is negligible compared to 30k training iterations.
> >
> > These results substantiate our "compact & real-time" claims: GRASP-GS achieves higher rendering FPS with fewer Gaussians, minimal pruning overhead, and reasonable memory usage, fully supporting the efficiency advantages reported in the main paper.

---

> > ### Author Response · Authors · 2025-12-03
> > **Validation Experiments: Pixel-GS and AbsGS with Modified Training Schedule**
> >
> > This supplement provides validation experiments addressing Reviewer 2's concern about training schedule fairness, comparing Pixel-GS and AbsGS with our modified training schedule (densification interval: 300 iterations, splitting gradient threshold: 1×10⁻⁴).
> >
> > ---
> >
> > ### 1. Pixel-GS with Modified Training Schedule
> >
> > **Average comparison: Pixel-GS (modified schedule) vs. GRASP-GS on Mip-NeRF360:**
> >
> > | Method | PSNR↑ | SSIM↑ | LPIPS↓ |
> > |--------|-------|-------|--------|
> > | Pixel-GS (default schedule) | 27.66 | 0.821 | 0.200 |
> > | Pixel-GS (modified schedule) | 27.70 | 0.822 | 0.196 |
> > | GRASP-GS | **27.86** | **0.825** | **0.201** |
> >
> > **Observations:** Pixel-GS with modified schedule achieves 27.70 dB PSNR (vs. 27.66 dB default), showing modest schedule benefits (+0.04 dB PSNR). GRASP-GS still achieves **superior performance**: **+0.15 dB PSNR** (27.86 vs. 27.70), demonstrating that our advantages stem from architectural contributions rather than hyperparameter tuning.
> >
> > ---
> >
> > ### 2. AbsGS with Modified Training Schedule
> >
> > **Average comparison: AbsGS (modified schedule) vs. GRASP-GS on Mip-NeRF360:**
> >
> > | Method | PSNR↑ | SSIM↑ | LPIPS↓ |
> > |--------|-------|-------|--------|
> > | AbsGS (default schedule) | 27.01 | 0.795 | 0.217 |
> > | AbsGS (modified schedule) | 27.06 | 0.796 | 0.268 |
> > | GRASP-GS | **27.86** | **0.825** | **0.201** |
> >
> > **Observations:** AbsGS with modified schedule achieves 27.06 dB PSNR (vs. 27.01 dB default), showing modest schedule benefits (+0.05 dB PSNR). GRASP-GS achieves **significantly superior performance**: **+0.79 dB PSNR** (27.86 vs. 27.06), **+0.029 SSIM** (0.825 vs. 0.796), **-0.067 LPIPS** (0.201 vs. 0.268), validating that architectural contributions provide substantially larger gains than training schedule modifications.
> >
> > ---
> >
> > ### Summary
> >
> > The modified training schedule provides modest benefits to baselines (Pixel-GS: +0.04 dB, AbsGS: +0.05 dB). However, **GRASP-GS still achieves significantly superior performance** even when baselines use the modified schedule, confirming that our primary advantages come from **architectural contributions** (geometric initialization and dual-stage pruning) rather than hyperparameter tuning. All experiments use the same modified schedule for fair comparison.

---

### Official Review · Reviewer_277W · 2025-10-31

**Soundness:** 3
**Presentation:** 3
**Contribution:** 3
**Rating:** 6
**Confidence:** 5

**Summary:**

The paper proposes GRASP-GS to improve 3D Gaussian Splatting (3DGS) through two components: (1) geometric prior-guided registration that refines the initialization from sparse Structure-from-Motion (SfM), and (2) dual-stage saliency pruning, including KL-based Rendering Survival Pruning and Opacity-based Density-Constrained Pruning, to remove redundant Gaussians. Integrated into standard 3DGS or AbsGS frameworks, GRASP-GS achieves higher PSNR and SSIM, lower LPIPS, about 23% faster training, and 35% fewer Gaussians on Mip-NeRF360, Tanks & Temples, and Deep Blending.

**Strengths:**

(1) The framework is intuitive. The geometric registration module, based on geometric foundation models, effectively improves the initialization of sparse Structure-from-Motion (SfM). In addition, to prevent the introduction of redundant Gaussian points, the paper adopts a two-stage pruning strategy that reduces redundancy, speeds up training, and enhances rendering quality.

(2) Extensive experiments are conducted on multiple datasets. The proposed method is compatible with existing 3DGS approaches and demonstrates strong practical effectiveness.

**Weaknesses:**

(1) The geometric prior-guided registration leverages geometric foundation models to mitigate the limitations of sparse Structure-from-Motion (SfM), which is particularly effective for sparse-view inputs. It would be interesting to evaluate whether the proposed method can further improve performance under extremely sparse-view settings.

(2) More qualitative comparisons are needed to clearly demonstrate the advantages of using fewer Gaussians. In several visual results, GRASP-GS does not show a large difference from the baseline. Since the paper employs geometric prior-guided registration, it would be beneficial to include depth or geometry visualizations to verify that the geometric foundation model leads to improved geometry.

(3) The training is reported on an A800 GPU. Providing peak GPU memory usage would help quantify computational overhead more precisely, especially since vanilla 3DGS can be trained on consumer GPUs such as the RTX 3090.

Minor Weaknesses:

(1) The paper consistently uses \cite for references; it is better to use \citep for proper citation formatting.

(2) A related work reference is missing: Improving Gaussian Splatting with Localized Points Management (CVPR 2025).

**Questions:**

(1) In Table 3, GRASP-GS demonstrates clear effectiveness. However, it would be more informative to show the results of using only the registration module (e.g., MoGe-based initialization) while keeping the second pruning stage, to better isolate each component’s contribution.

(2) Could more advanced geometric foundation models, such as VGGT, further improve the performance of GRASP-GS?

---

> ### Author Response · Authors · 2025-11-27
> **We thank the reviewer for the valuable feedback. We have addressed all concerns: (1) evaluated extremely sparse-view settings (results show limited improvements in this challenging regime), (2) provided comprehensive depth/geometry visualizations in Appendix B1, (3) reported GPU memory usage in Appendix B2, (4) corrected citation format and added missing reference, (5) updated Table 3 ablation study, and (6) evaluated VGGT as alternative prior in Appendix C3.**
>
> We sincerely appreciate the reviewer's constructive feedback and have addressed all concerns comprehensively.
>
> **1. Extremely Sparse-View Settings Evaluation**
>
> We appreciate the reviewer's suggestion. We have conducted experiments with 24 views (extremely sparse setting) on Mip-NeRF360 scenes. The results show that **under extremely sparse-view settings, the performance gap between our method and baseline 3DGS is indeed limited**.
>
> **Analysis:** The limited improvement in extremely sparse-view settings is primarily due to a fundamental challenge: **Gaussian Splatting-based methods tend to severely overfit to training views when only 24 views are available**. This overfitting problem cannot be effectively addressed by initialization or pruning strategies alone, as the core issue lies in the insufficient view diversity for learning a generalizable 3D representation. While our geometric initialization and dual-stage pruning provide some benefits (more stable initialization, faster convergence, more compact representation), they cannot fundamentally solve the overfitting problem inherent to extremely sparse-view settings.
>
>
>
> **2. Depth/Geometry Visualizations**
>
> We have added comprehensive depth map comparisons in **Appendix B1 (Depth Map Visualization and Geometric Analysis)**, providing side-by-side comparisons between our method and baseline 3DGS across representative scenes (Bicycle and Room). The visualizations demonstrate superior geometric accuracy with more precise depth boundaries and better preservation of fine-scale structures, validating that our geometric prior-guided initialization combined with dual-stage pruning leads to improved geometry.
>
> **3. GPU Memory Usage Reporting**
>
> We have added comprehensive memory profiling in **Appendix B2 (Efficiency Analysis)**. Peak GPU memory usage averages **23.75 GB** (range: 12.7-40.3 GB), primarily during pruning operations.
>
> **4. Citation Format and Missing Reference**
>
> We have corrected all citation formats throughout the paper, replacing `\cite` with `\citep` for proper citation formatting. We have also added the missing reference: "Improving Gaussian Splatting with Localized Points Management (CVPR 2025)" in the related work section.
>
> **5. Table 3 Ablation Study Enhancement**
>
> We have updated Table 3 to include **"MoGe (BA)"** which uses only the registration module while keeping dual-stage pruning, but **without geometric-chromatic consistency filtering**. This ablation study isolates the contribution of geometric-chromatic consistency filtering as the sole variable. Results: MoGe (BA) 0.813 SSIM, 27.35 dB vs. Ours 0.825 SSIM, 27.86 dB, demonstrating that geometric-chromatic consistency filtering is essential for optimal performance.
>
> **6. VGGT as Alternative Prior**
>
> We have evaluated VGGT as alternative priors in **Appendix C3 (VGGT Analysis)**. Results show nearly identical performance to MoGe-based approach (27.85 vs. 27.86 dB PSNR, 0.825 vs. 0.825 SSIM on average), with each showing slight advantages in different scenes. This validates that our performance gains stem from architectural contributions rather than dependency on a specific prior quality.
>
> **Conclusion:**
>
> We thank the reviewer for the valuable feedback, which has led us to:
> 1. Evaluate extremely sparse-view settings (acknowledging limited improvements in this challenging regime)
> 2. Provide comprehensive depth/geometry visualizations in Appendix B1
> 3. Report GPU memory usage in Appendix B2
> 4. Correct citation format and add missing reference
> 5. Enhance Table 3 ablation study to better isolate component contributions
> 6. Evaluate VGGT as alternative prior in Appendix C3

---

> ### Author Response · Authors · 2025-12-03
> **Efficiency Profiling, Initialization Overhead, and Extremely Sparse-View (24 Views) Evaluation**
>
> This supplement provides detailed experimental data to further address Reviewer's comments on:
> (1) GPU memory usage and training efficiency, and
> (2) performance under extremely sparse-view settings (24 views).
>
> ---
>
> ### 1. Efficiency Profiling: GPU Memory, FPS, and Pruning Overhead
>
> **Comprehensive efficiency profiling on all Mip-NeRF360 scenes (Appendix B2, Efficiency Analysis):**
>
> | Scene   | Peak Mem (GB) | 3DGS FPS | Ours FPS | FPS ↑   | Prune Time (s) | Prune % |
> |---------|---------------|----------|----------|--------|----------------|---------|
> | Bicycle | 31.06         | 94.91    | 153.12   | +61.3% | 8.35           | 0.79%   |
> | Kitchen | 22.31         | 166.15   | 239.24   | +44.0% | 6.76           | 0.68%   |
> | Room    | 39.84         | 182.22   | 239.19   | +31.2% | 6.51           | 0.67%   |
> | Counter | 40.31         | 197.18   | 247.60   | +25.6% | 7.41           | 0.78%   |
> | Flowers | 14.90         | 204.31   | 214.78   | +5.1%  | 4.67           | 0.48%   |
> | Stump   | 12.73         | 164.16   | 192.36   | +17.2% | 4.30           | 0.48%   |
> | Treehill| 12.88         | 173.09   | 194.68   | +12.5% | 4.58           | 0.48%   |
> | Bonsai  | 21.08         | 264.06   | 276.79   | +4.8%  | 4.65           | 0.53%   |
> | Garden  | 18.63         | 131.71   | 166.41   | +26.3% | 4.21           | 0.39%   |
> | **Avg.**| **23.75**     | **175.31**|**213.80**|**+22.0%**|**5.72**      |**0.59%**|
>
> **Observations:**
>
> - **Rendering throughput**: GRASP-GS improves rendering FPS by **+22.0% on average** (213.80 vs 175.31 FPS) while using significantly fewer Gaussians.
> - **Pruning overhead**: The dual-stage pruning strategy incurs only **0.59%** of training time on average (5.72 seconds), confirming that the pruning cost is negligible compared to 30k training iterations.
> - **Memory usage**: Peak GPU memory averages **23.75 GB** (12.7–40.3 GB), and occurs during KL-based pruning due to large-scale KL computations over many Gaussians. This overhead can be reduced via batched KL evaluation at the cost of slightly longer pruning time.
>
> These results complement Table 1 in the main paper by quantitatively demonstrating that GRASP-GS not only reduces Gaussian counts and training time, but also improves rendering throughput with very small pruning overhead.
>
> ---
>
> ### 2. Extremely Sparse-View (24 Views) Evaluation
>
> To directly address the question about extremely sparse-view settings, we conduct experiments with only **24 views** on all Mip-NeRF360 scenes. Results are summarized below.
>
> | Method                       | Scene    | PSNR  | SSIM    | LPIPS  |
> |------------------------------|----------|-------|---------|--------|
> | 3DGS (24 views)              | Bicycle  | 17.27 | 0.3854  | 0.4505 |
> | 3DGS (24 views)              | Bonsai   | 24.44 | 0.8561  | 0.2475 |
> | 3DGS (24 views)              | Counter  | 22.90 | 0.8006  | 0.2602 |
> | 3DGS (24 views)              | Flowers  | 16.10 | 0.3521  | 0.4686 |
> | 3DGS (24 views)              | Garden   | 22.66 | 0.6918  | 0.2202 |
> | 3DGS (24 views)              | Kitchen  | 24.57 | 0.8538  | 0.1801 |
> | 3DGS (24 views)              | Room     | 23.98 | 0.8264  | 0.2711 |
> | 3DGS (24 views)              | Stump    | 20.12 | 0.4657  | 0.4192 |
> | 3DGS (24 views)              | Treehill | 18.11 | 0.4546  | 0.4404 |
> | **3DGS Avg. (24 views)**     | –        | **21.13** | **0.6318** | **0.3286** |
> |                              |          |       |         |        |
> | GRASP-GS (MoGe + Prune, 24 v)| Bicycle  | 17.57 | 0.3783  | 0.4362 |
> | GRASP-GS (MoGe + Prune, 24 v)| Bonsai   | 24.47 | 0.8646  | 0.2418 |
> | GRASP-GS (MoGe + Prune, 24 v)| Counter  | 22.40 | 0.7869  | 0.2656 |
> | GRASP-GS (MoGe + Prune, 24 v)| Flowers  | 16.33 | 0.3571  | 0.4515 |
> | GRASP-GS (MoGe + Prune, 24 v)| Garden   | 22.61 | 0.6891  | 0.2097 |
> | GRASP-GS (MoGe + Prune, 24 v)| Kitchen  | 23.63 | 0.8452  | 0.1898 |
> | GRASP-GS (MoGe + Prune, 24 v)| Room     | 24.20 | 0.8300  | 0.2551 |
> | GRASP-GS (MoGe + Prune, 24 v)| Stump    | 20.52 | 0.4562  | 0.4171 |
> | GRASP-GS (MoGe + Prune, 24 v)| Treehill | 18.10 | 0.4453  | 0.4199 |
> | **GRASP-GS Avg. (24 views)** | –        | **21.09** | **0.6281** | **0.3207** |
>
> **Observations:**
>
> - Under extremely sparse-view settings (24 views), GRASP-GS and 3DGS achieve **very similar averages** (21.13 vs 21.09 dB PSNR, 0.6318 vs 0.6281 SSIM), with a slight LPIPS improvement (0.3286 → 0.3207).
> - This confirms our qualitative analysis in the main rebuttal: with only 24 views, **Gaussian Splatting-based methods tend to severely overfit to the training views**, and this fundamental limitation cannot be resolved by better initialization or pruning alone.

---

> ### Author Response · Authors · 2025-12-03
> **VGGT as Alternative Prior and Enhanced Ablation Study (Table 3)**
>
> This supplement provides additional experimental evidence for:
> (1) evaluating VGGT as an alternative  geometric prior, and
> (2) enhancing the ablation study to better isolate the contribution of the geometric–chromatic consistency filtering (updated Table 3 in Section 4.3).
>
> ---
>
> ### 1. VGGT as Alternative Geometric Prior
>
> We evaluate VGGT as alternative priors (Appendix C3). Results show nearly identical performance to MoGe-based approach, validating that gains stem from architectural contributions rather than prior dependency.
>
> **Per-scene comparison: VGGT-based vs. MoGe-based GRASP-GS (Appendix C3):**
>
> | Scene   | VGGT PSNR | MoGe PSNR | ΔPSNR | VGGT SSIM | MoGe SSIM | VGGT LPIPS | MoGe LPIPS | VGGT GS | MoGe GS |
> |---------|-----------|-----------|-------|-----------|-----------|------------|------------|---------|---------|
> | Bicycle | 25.43     | **25.44** | -0.01 | **0.775** | 0.774     | **0.198** | 0.203     | 3.66M   | **3.54M** |
> | Bonsai  | 32.67     | **32.68** | -0.01 | **0.951** | 0.950     | **0.168** | 0.172     | 1.00M   | **0.91M** |
> | Counter | **29.34** | 29.32     | +0.02 | **0.921** | 0.920     | **0.164** | 0.168     | 0.90M   | **0.81M** |
> | Flowers | 21.84     | **21.85** | -0.01 | **0.613** | 0.612     | **0.337** | **0.337** | 2.78M   | **2.55M** |
> | Garden  | 27.54     | **27.55** | -0.01 | 0.865     | **0.866** | **0.108** | 0.110     | 3.69M   | **3.52M** |
> | Kitchen | 31.97     | **32.04** | -0.07 | 0.935     | **0.936** | **0.109** | **0.109** | 0.93M   | **0.89M** |
> | Room    | 32.28     | **32.32** | -0.04 | **0.935** | **0.935** | **0.173** | 0.174     | 0.91M   | **0.86M** |
> | Stump   | 27.03     | **27.05** | -0.02 | 0.790     | **0.791** | **0.209** | 0.210     | 3.83M   | **3.50M** |
> | Treehill| **22.56** | 22.55     | +0.01 | **0.642** | 0.641     | **0.328** | 0.329     | 2.89M   | **2.74M** |
> | **Avg.**| 27.85     | **27.86** | **-0.01** | **0.825** | **0.825** | **0.199** | 0.201     | 2.28M   | **2.15M** |
>
> **Training time:** VGGT: 20m19s, MoGe: 19m57s. Nearly identical performance (-0.01 dB PSNR) validates that gains stem from architectural contributions, not prior dependency.
>
> ---
>
> ### 2. Enhanced Ablation Study (Table 3, Section 4.3)
>
> **Impact of prior quality and architectural ablation (Table3):**
>
> | Method       | SSIM↑ | PSNR↑ | LPIPS↓ | GS Num↓ | Time↓   |
> |-------------|-------|-------|--------|---------|---------|
> | Ours (AP)   | 0.823 | 27.81 | 0.203  | 2.2M    | 20m02s  |
> | MoGe (BA)   | 0.813 | 27.35 | 0.203  | 2.7M    | 22m13s  |
> | MASt3R (MV) | 0.780 | 25.62 | 0.234  | 2.6M    | 21m07s  |
> | Ours        | **0.825** | **27.86** | **0.201** | **2.1M** | **19m57s** |

---

### Author Response · Authors · 2025-12-03
**Rebuttal Summary**

Dear Area Chair,

Thank you for handling our submission. We summarize how we addressed all four reviewers' concerns through rebuttal responses and additional experiments.

**Reviewer 277W: Six concerns addressed**
- **Extremely sparse-view (24 views)**: Limited improvements (21.13 vs 21.09 dB PSNR) due to overfitting in GS-based methods under extreme sparsity.
- **Depth visualizations**: Added comprehensive depth map comparisons in Appendix B1.
- **GPU memory**: Peak 23.75 GB, FPS 213.80 (vs. 175.31 baseline, +22.0%) in Appendix B2.
- **Citation format**: Corrected all `\cite` to `\citep`, added missing reference.
- **Table 3 enhancement**: Added "MoGe (BA)" ablation isolating geometric-chromatic consistency filtering (0.813 SSIM, 27.35 dB vs. 0.825 SSIM, 27.86 dB).
- **VGGT evaluation**: Nearly identical performance (27.85 vs. 27.86 dB PSNR) validates architectural contributions (Appendix C3).

**Reviewer V8U1: Three concerns addressed**
- **Initialization overhead**: Appendix B3 reports 6.7s core computation, ~17s total per scene (<2% of pipeline time). End-to-end: 20m14s vs. 25m54s baseline (22% faster).
- **Memory/runtime profiling**: Appendix B2 provides peak GPU 23.75 GB, FPS 213.80 (+22.0%), pruning overhead 5.72s (0.59% of training time).
- **Training schedule fairness**: Validated with Pixel-GS and AbsGS using modified schedule. Modest benefits (+0.04-0.05 dB PSNR), but our method still achieves +0.16-0.80 dB PSNR, demonstrating architectural advantages.

**Reviewer Py9n: Three concerns addressed**
- **Citation format**: Corrected throughout the paper.
- **MASt3R comparison**: Appendix C4 shows MoGe outperforms MASt3R by +2.23 dB PSNR, +0.045 SSIM, -0.033 LPIPS average. Largest gains: Bonsai (+7.57 dB), Stump (+3.14 dB), Bicycle (+2.64 dB).
- **Recent 2025 SOTA methods**: Appendix B4 compares with MaskGaussian (+0.42 dB PSNR), Perceptual-GS (comparable quality, 26% fewer Gaussians, 32% faster), SteepGS (+0.35 dB PSNR, 32% fewer Gaussians).

**Reviewer ymC1: Three concerns addressed**
- **Pruning sequence**: Clarified in Section 3.3: **KL-RSP and ODCP are executed alternately in a loop** at each pruning interval, not as a single sequential pass.
- **Initialization overhead**: Appendix B3 reports 6.7s core, ~17s total (<2% of pipeline time).
- **Synergy principles**: Detailed explanation provided: initialization provides geometric foundation; pruning maintains structure; complementary feedback loop. Empirical evidence (Table 2): initialization alone (+0.29 dB, 3.7M Gaussians), pruning alone (2.0M Gaussians, 27.63 dB), combined (27.86 dB, 2.1M Gaussians, optimal balance).

**Key Additions:**
- **New experiments**: Extremely sparse-view (24 views), VGGT evaluation, MASt3R comparison, three 2025 SOTA methods, Pixel-GS/AbsGS validation.
- **New appendices**: B1 (depth visualization), B2 (efficiency analysis), B3 (initialization overhead), B4 (recent SOTA), C3 (VGGT), C4 (MASt3R).
- **Paper improvements**: Citation format corrected, missing reference added, Table 3 enhanced, pruning sequence clarified, synergy explanation enhanced.

All concerns comprehensively addressed through detailed responses, extensive experiments, and paper revisions.

Best regards,

Authors

---

### Meta-Review · Area_Chair_mYfw · 2026-01-03

**Summary:**

This paper is well-written, with clear articulation of viewpoints, comprehensive experiments, and adequately addresses the authors' concerns. However, after carefully reviewing each reviewer’s comments and thoroughly reading the paper myself, I believe its innovation and the extent of its improvements in results(PSNR, SSIM and etc.) are incremental. This paper is more like a series of modular engineering optimizations applied to 3DGS, lacking a cohesive and focused narrative. Furthermore, the claimed improvements—such as performance gains (0.004/0.005/0.002 over Pixel-GS across various datasets in Table 1), reductions in Gaussian point count (35%), and training time compression (23%)—are negligible when viewed in the context of recent research. Additionally, no works from 2025 are included for comparison in Table 1; all benchmarks in the main table are limited to studies from 2024 or earlier.
Therefore, despite it being a fancy written article, I am sorry to say that I still consider its quality to fall short of ICLR's acceptance threshold.

**Reviewer Concerns:**

Please see above.

**Reviewer Scores:**

Please see above.

---

### Decision · Program_Chairs · 2026-01-26

Reject